# From Weather Data to River Runoff: Using Spatiotemporal Convolutional Networks for Discharge Forecasting

Florian Börgel[*,1], Sven Karsten[1], Karoline Rummel[1], and Ulf Gräwe[1]

[1]Leibniz-Institute for Baltic Sea Research Warnemünde, Rostock, Germany

**Correspondence:** Florian Börgel (florian.boergel@io-warnemuende.de)

**Abstract.** The quality of the river runoff determines the quality of regional climate projections for coastal oceans or other estuaries. This study presents a novel approach to river runoff forecasting using Convolutional Long Short-Term Memory (ConvLSTM) networks. Our method accurately predicts daily runoff for 97 rivers within the Baltic Sea catchment by modeling runoff as a spatiotemporal sequence defined by atmospheric forcing. The ConvLSTM model predicts river runoff with an accuracy of $\pm 5\%$ when compared to the hydrological model. Compared to more complex process-based hydrological models, ConvLSTM offers fast processing times and easy integration into climate models, demonstrating its potential as a powerful tool for climate simulation and water resource management.

## 1 Introduction

River runoff is a key component of the global water cycle as it comprises about one third of the precipitation over land areas (Hagemann et al., 2020), making accurate runoff forecasting essential for effective water resources management, particularly over extended periods (Fang et al., 2019; Tan et al., 2018). In addition to short-term forecasting, long-term projections of river runoff are vital for climate change studies, projecting flooding and droughts over global and river basins (Cook et al., 2020). These studies calculate river runoff using a land model incorporating a hydrological model within a coupled earth system model (ESM) (Wang et al., 2022). In the absence of a fully coupled ESM, river runoff as input for ocean models can be created using hydrological models such as the Hydrological Discharge (HD) model (Hagemann et al., 2020) or the HYdrological Predictions for the Environment (HYPE) model (Lindström et al., 2010). Hydrological models represent a process-based approach, where the water balance is calculated using hydrological processes (e.g., snow, glaciers, soil moisture, groundwater contribution). These models are complex forecasting tools widely utilized, such as high-resolution multi-basin models applied across Europe (Hundecha et al., 2016).

The second approach to projecting river runoff employs data-driven models, such as calculating river runoff as the difference between precipitation and evaporation over a catchment area, with an integrated statistical correction (Meier et al., 2012). With the recent rise of machine learning in climate research, various data-driven model architectures have been explored for river runoff forecasting. Common approaches include feed-forward artificial neural networks, support vector machines, adaptive neuro-fuzzy inference systems, and, notably, Long Short-Term Memory (LSTM) neural networks. LSTMs have gained traction for long-term hydrological forecasting due to their excellent performance (Humphrey et al., 2016; Huang et al., 2014; Ashrafi

et al., 2017; Liu et al., 2020; Fang and Shao, 2022; Kratzert et al., 2018). LSTM networks, first introduced by Hochreiter and Schmidhuber (1997), are an evolution of the classical Recurrent Neural Networks (Sherstinsky, 2020). A significant advantage of the LSTM's architecture is the memory cell's ability to retain gradients. This mechanism addresses the vanishing gradient problem, where, as input sequences elongate, the influence of initial stages becomes harder to capture, causing gradients of early input points to approach zero. LSTMs have shown stability and efficacy in sequence-to-sequence predictions. However, a limitation of LSTMs is their inability to effectively capture two-dimensional structures, an area where Convolutional Neural Networks (CNNs) excel (Höhlein et al., 2020). Recognizing this limitation, Shi et al. (2015) proposed a Convolutional LSTM (ConvLSTM) architecture, which combines the strengths of both LSTM and CNN. In practical applications, the combination of LSTMs and CNNs in the form of ConvLSTM models allowed for improving the accuracy of precipitation nowcasting (Shi et al., 2015), flood forecasting (Moishin et al., 2021), and river runoff forecasting (Ha et al., 2021; Zhu et al., 2023).

We use the Baltic Sea catchment as an example to illustrate our approach. Although the methodology we propose is universally applicable across various geographic regions, the Baltic Sea represents a challenging region due to its unique hydrological characteristics, nearly decoupled from the open ocean (see Figure 1) (Meier and Döscher, 2002). Freshwater enters the Baltic Sea through river runoff or positive net precipitation (precipitation minus evaporation) over the sea surface. The net precipitation accounts for 11% and the river input for 89% of the total freshwater input (Meier and Döscher, 2002). Consequently, the Baltic Sea's sea surface salinity (SSS) is largely driven by freshwater supply from rivers.

SSS is an essential variable for the marine ecosystem in the Baltic Sea, as most species are adapted to either marine or freshwater conditions. Therefore, biases in the SSS's spatial and temporal variability significantly impact primary production and fish biomass (Kniebusch et al., 2019). Accurate modeling of the Baltic Sea heavily relies on the quality of the river input data used in simulations. Analyzing nearly 100 years of observations, (Winsor et al., 2001) found that variations in freshwater storage are closely correlated with accumulated changes in river runoff. From 1902 to 1998, the average freshwater inflow to the Baltic amounted to 16,115 $m^3$/s, with contributions from river runoff (14,085 m$^3$/s) and net precipitation over the Baltic Sea (2,030 m$^3$/s) (Meier and Kauker, 2003). This freshwater inflow results in a residence time of about 35 years for freshwater in the Baltic Sea (Omstedt and Hansson, 2006; Winsor et al., 2001; Meier and Kauker, 2003).

In this work, we demonstrate that ConvLSTM networks are a reliable method for predicting multiple rivers simultaneously, using only atmospheric forcing as input data, which allows us to emulate a hydrological model. The main focus of this work is to present a ConvLSTM architecture capable of predicting daily river runoff for 97 rivers across the Baltic Sea catchment. To train the network, we use data from the E-HYPE model (Väli et al., 2019) as reference output data and data from the UERRA project (Uncertainties in Ensembles of Regional Reanalyses, http://www.uerra.eu/) as atmospheric forcing (Section 3). The quality of the model is evaluated in Section 4. The obtained results are further discussed and concluded in Section 5.

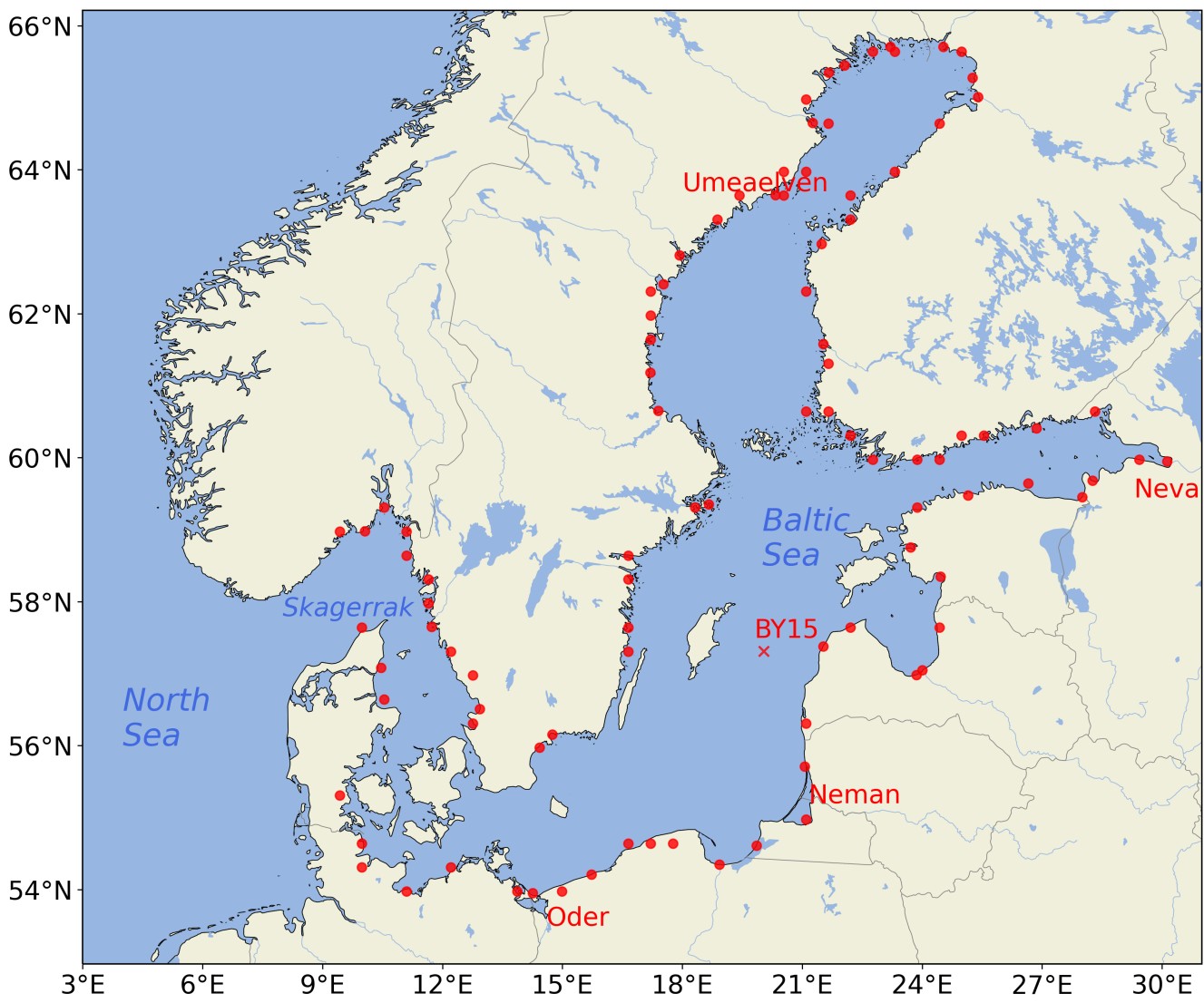

**Figure 1.** Map of the Baltic Sea region. The catchment area is indicated by hatched regions. Red dots represent the locations of major rivers that flow into the Baltic Sea as represented in the hydrological model E-HYPE. The annotation BY15 marks the chosen validation station situated in the central Baltic Sea, used for validating the regional ocean model. The four remaining annotations (red) indicate the positions of the rivers that will be evaluated in detail.

## 2 Implemented model architecture

### 2.1 The main idea

We assume that the runoff at a specific point in time $t$ for all $N_r$ considered rivers collected in the vector $\boldsymbol{R}^t \in \mathbb{R}^{N_r}$, can be accurately approximated by a functional $\boldsymbol{M}(\{X^{t_k}[x,y,\tau]\})$ of $k = 1, \ldots, N_k$ atmospheric fields $X^{t_k}[x,y,\tau]$ which are known for the past $\tau = 1, \ldots, N_\tau$ time instances. This relationship is expressed as:

$$\boldsymbol{R}^t = \boldsymbol{M}(\{X_k^t[x,y,\tau]\}) \,. \tag{1}$$

The atmospheric fields are evaluated over a spatial domain $x = 1, \ldots N_x$ and $y = 1, \ldots N_y$, which is sufficiently large to capture all significant local and non-local contributions of the atmospheric fields to the river runoff. Typically, such a mapping is realized using a hydrological model that simulates all relevant physical processes, transforming variables like precipitation and evaporation into river runoff. This process relies heavily on domain knowledge to tune all parameters to reasonable values. As an alternative, combining a convolutional Long-Short Term Memory (ConvLSTM) model with a subsequent fully connected (FC) neural network can adequately represent this functional. This approach eliminates the need for detailed knowledge of the involved processes and their modeling. Instead, these features can be "learned" by the network in an automated manner, i.e., all free parameters are optimized such that the network's output reproduces the data of the hydrological model with given atmospheric input fields. Our proposed network architecture is visualized in Figure 2 and described in detail in the following sections. To provide an overview, we will discuss the main components of this architecture one by one.

### 2.2 The ConvLSTM network

#### 2.2.1 The LSTM approach

Before turning directly to the ConvLSTM, the simpler architecture of the plain Long-Short Term Memory (LSTM) model is examined. This serves as a foundation for understanding the more complex ConvLSTM. The LSTM, a specialized form of Recurrent Neural Networks (RNNs), is specifically designed to model temporal sequences $\boldsymbol{X}^t[1], \ldots \boldsymbol{X}^t[\tau], \ldots \boldsymbol{X}^t[N_\tau]$ of $N_\tau$ input quantities $\boldsymbol{X}^t[\tau] = (X_k^t[\tau]) \in \mathbb{R}^{N_k}$. This sequence is taken from a dataset given in form of a time series $\{\boldsymbol{X}^t\}$ with the endpoint coinciding with the specific element in the time series connected to time $t$, i.e. $\boldsymbol{X}^t[N_\tau] \equiv \boldsymbol{X}^t$, see Figure 2. Here, $N_k$ represents the number of input "channels," which can correspond to different measurable quantities. The LSTM's unique design allows it to adeptly handle long-range dependencies, setting it apart from traditional RNNs in terms of accuracy (see Figure 3).

The critical component of the LSTM's innovation is its cell state, $\boldsymbol{C}^t[\tau] = (C_h^t[\tau]) \in \mathbb{R}^{N_h}$, which stores state information, also referred to as long-term memory. This state information complements the so-called hidden state vector $\boldsymbol{H}^t[\tau] = (H_h^t[\tau]) \in \mathbb{R}^{N_h}$, which is also known from simpler neural network architectures. In the case of the LSTM, the hidden state vector plays the role of the short-term memory. The cell state and the hidden state are vectors, where each element is associated with one of the

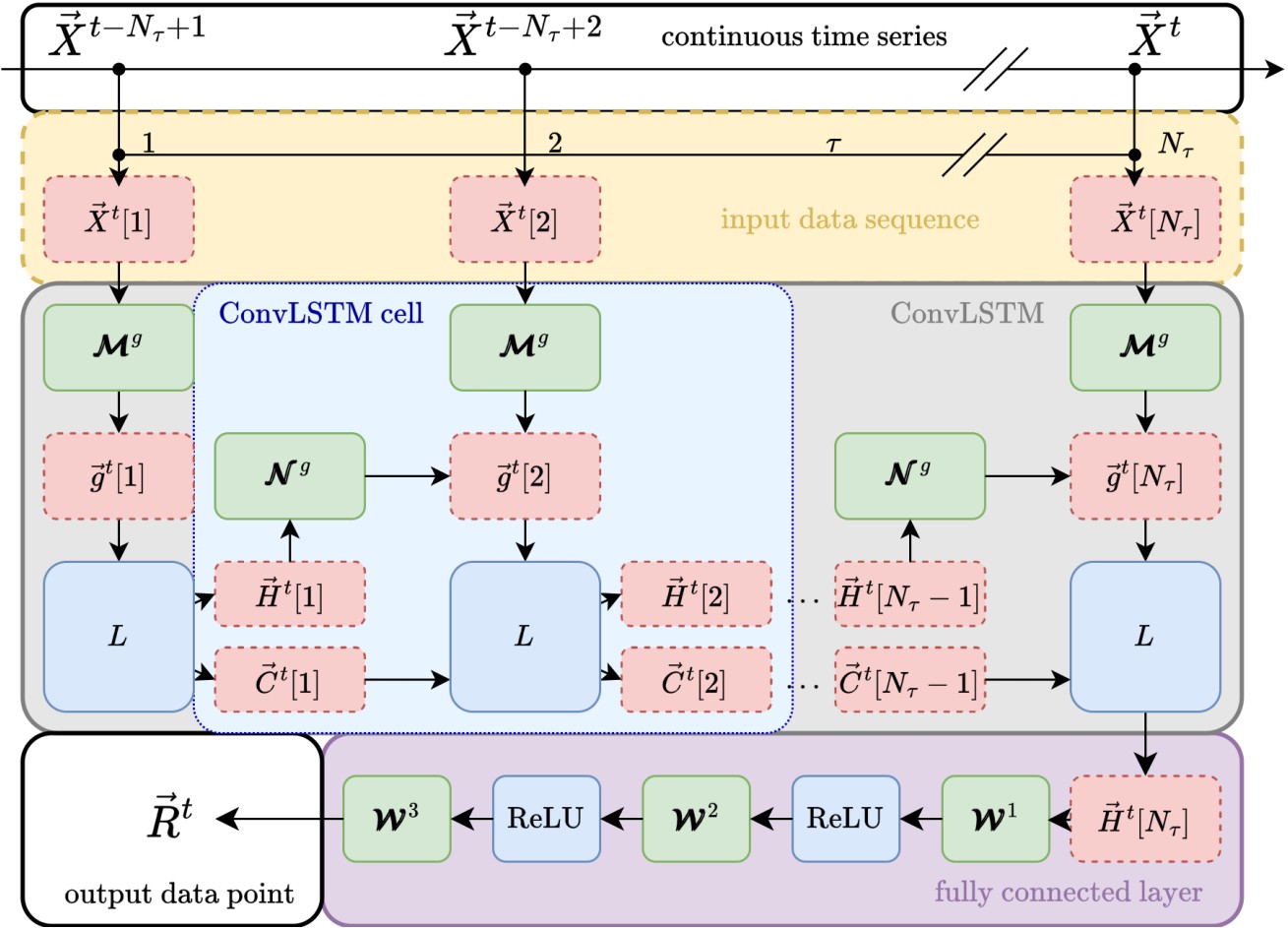

**Figure 2.** Combined ConvLSTM and FC network architecture. The starting point is the continuous time series of input data $\{\boldsymbol{X}^t\}$, upper white block. From this series, a contiguous sequence of $N_\tau$ elements (yellow block) is used to feed a chain of $N_\tau$ connected ConvLSTM cells (light blue block) building the ConvLSTM network (grey block). The input sequence is mapped via weighting matrices $\boldsymbol{\mathcal{M}}^g$ (green blocks) onto gate vectors $\boldsymbol{g}^t[\tau]$. The gate vectors are then used to update the cell state $\boldsymbol{C}^t[\tau-1]$ and the hidden state $\boldsymbol{H}^t[\tau-1]$ of the last ConvLSTM cell to the current values $\boldsymbol{C}^t[\tau]$ and $\boldsymbol{H}^t[\tau]$, respectively. The update is performed with the LSTM core equations collectively described by the mapping $L$, see (4) The weighting matrices $\boldsymbol{\mathcal{N}}^g$ (green blocks) control how much of the last hidden state enters the updated state. The final output of the ConvLSTM $\boldsymbol{H}^t[\tau]$ is then propagated to a FC network, which itself is a chain of three FC layers consisting of weighting matrices $\boldsymbol{\mathcal{W}}$ and connected via ReLU functions, see Section 2.4. The final result is the river runoff $\boldsymbol{R}^t$ for all rivers considered at the current time instance $t$ (white block on the lower left). Note that all bias vectors are omitted for the sake of clarity. See text for more information.

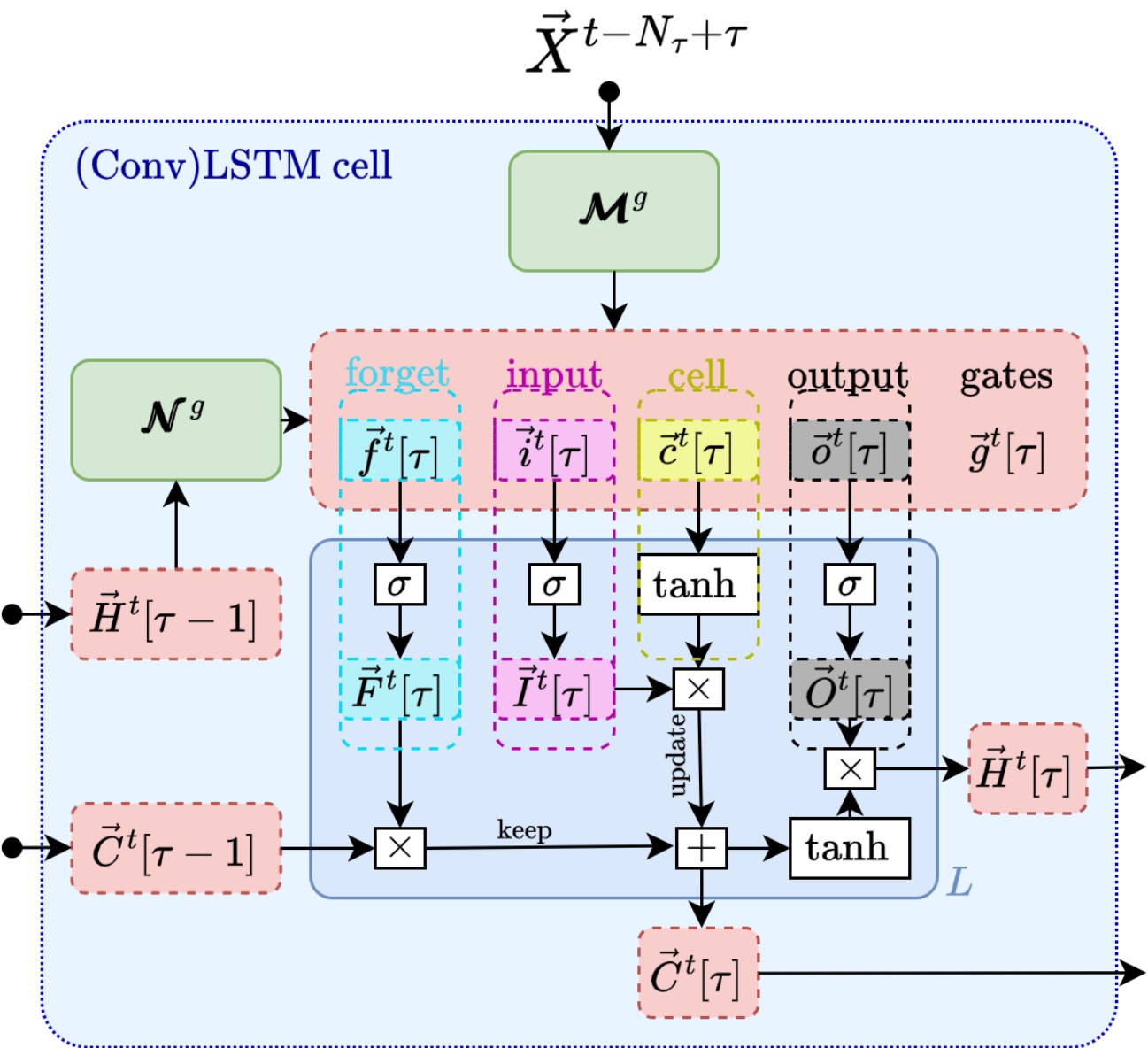

**Figure 3.** Inner structure of a Long Short-Term Memory Cell. See Figure 2 and text for information.

$N_h$ hidden layers, labeled by $h$. These internal, artificial degrees of freedom enable the high adaptability of neural networks. The two state vectors are determined through several self-parameterized gates, all in the same vector space as $\boldsymbol{C}^t[\tau]$, see Figure 3 for a visualization.

In particular, the forget gate $\boldsymbol{F}^t[\tau]$ defines the portion of the previous (long-term memory) cell state $\boldsymbol{C}^t[\tau-1]$ that should be kept, see dashed cyan box therein. The input gate $\boldsymbol{I}^t[\tau]$ controls the contribution of the current input used to update the long-term memory, $\boldsymbol{C}^t[\tau]$ (magenta and yellow boxes). The output gate, $\boldsymbol{O}^t[\tau]$, then determines how much of this updated long-term memory contributes to the new (short-term memory) hidden state, $\boldsymbol{H}^t[\tau]$ (black dashed box).

For a fixed point $\tau$ in the sequence, the action of an LSTM cell, i.e. the connection between the input $\boldsymbol{X}^t[\tau]$, the various gates and the state vectors, is mathematically given as follows. First, the elements of the input sequence together with the hidden state are mapped onto auxiliary gate vectors, collectively denoted by $\boldsymbol{g}^t[\tau] = (g_h^t[\tau]) \in \mathbb{R}^{N_h}$, via

$$g_h^t[\tau] = \mathcal{M}_{hk}^g X_k^t[\tau] + \mathcal{N}_{hh'}^g H_{h'}^t[\tau-1] + \mathcal{B}_h^g \,, \tag{2}$$

where $g \in i, f, o, c$ stands for the input, forget, output, and cell-state gate, respectively, and Einstein's summation convention is employed, i.e., indices that appear twice are summed over. The calligraphic symbols $\mathcal{M}_{hk}^g, \mathcal{N}_{hh'}^g$ and $\mathcal{B}_h^g$ are the free parameters of the network that are optimized for the given problem during the training, which is at the heart of any machine learning approach. The matrix $\boldsymbol{\mathcal{M}}^g = (\mathcal{M}_{hk}^g) \in \mathbb{R}^{N_h \times N_k}$ can be interpreted as a Markovian-like contribution of the current input $\boldsymbol{X}^t[\tau]$ to the gates, whereas the $\boldsymbol{\mathcal{N}}^g = (\mathcal{N}_{hh'}^g) \in \mathbb{R}^{N_h \times N_h}$ scales a non-Markovian part determined by the hidden state of the last sequence point $\tau-1$. The vector $\boldsymbol{\mathcal{B}}^g = (\mathcal{B}_h^g) \in \mathbb{R}^{N_h}$ is a learnable bias. It should be stressed that these parameters do neither depend on $t$ nor on $\tau$ and are thus optimized once for the complete dataset $\{\boldsymbol{X}^t\}$.

Note that this mapping is sometimes extended by a contribution to the $g_h^t[\tau]$ from the past cell state $\boldsymbol{C}^t[\tau-1]$. Nevertheless, this "peeping" mechanism is not further considered in this work. For the sake of brevity, we can write the mapping more compactly in matrix-vector form as

$$\boldsymbol{g}^t[\tau] = \boldsymbol{\mathcal{M}}^g \boldsymbol{X}^t[\tau] + \boldsymbol{\mathcal{N}}^g \boldsymbol{H}^t[\tau-1] + \boldsymbol{\mathcal{B}}^g \tag{3}$$

Second, the actual gate vectors are computed by the core equations of the LSTM as proposed by (Hochreiter and Schmidhuber, 1997):

$$
\begin{aligned}
\boldsymbol{I}^t[\tau] &= \sigma(\boldsymbol{i}^t[\tau]) \\
\boldsymbol{F}^t[\tau] &= \sigma(\boldsymbol{f}^t[\tau]) \\
\boldsymbol{O}^t[\tau] &= \sigma(\boldsymbol{o}^t[\tau]) \\
\boldsymbol{C}^t[\tau] &= \boldsymbol{F}^t[\tau] \circ \boldsymbol{C}^t[\tau-1] + \boldsymbol{I}^t[\tau] \circ \tanh(\boldsymbol{c}^t[\tau]) \\
\boldsymbol{H}^t[\tau] &= \boldsymbol{O}^t[\tau] \circ \tanh(\boldsymbol{C}^t[\tau]) \,,
\end{aligned}
\tag{4}
$$

where $\sigma$ denotes the logistic sigmoid function, $\tanh$ is the hyperbolic tangent, and $\circ$ stands for the Hadamard product (all applied in an element-wise fashion to the vectors). In the last two equations, the role of the input, forget, and output gates as described above, becomes apparent.

The third step in a single layer LSTM (as employed for the work presented here) is to provide the output of the current LSTM cell, i.e. $\boldsymbol{H}^t[\tau]$ and $\boldsymbol{C}^t[\tau]$, to the subsequent LSTM cell that processes the next element $\boldsymbol{X}^t[\tau+1]$ of the input sequence.

The full action of the LSTM network up to the end of the sequence can be written as a nested function call,

$$\left(\boldsymbol{H}^t[N_\tau], \boldsymbol{C}^t[N_\tau]\right) = L\left(\boldsymbol{X}^t[N_\tau], L\left(\boldsymbol{X}^t[N_\tau-1], \ldots L\left(\boldsymbol{X}^t[1], (\boldsymbol{H}^t[0], \boldsymbol{C}^t[0])\right) \ldots\right)\right) , \tag{5}$$

where $L\left(\boldsymbol{X}^t[\tau], (\boldsymbol{H}^t[\tau-1], \boldsymbol{C}^t[\tau-1])\right)$ represents (3) and (4). For the present work, the initial conditions are chosen as $\boldsymbol{H}^t[0] = \boldsymbol{C}^t[0] = 0$, which means that there is no memory longer than $N_\tau$ time steps.

The final output of the ConvLSTM chain, $\boldsymbol{H}^t[N_\tau]$ and $\boldsymbol{C}^t[N_\tau]$, encode information on the entire input sequence ending at time $t$. This information must be decoded via an appropriate subsequent network, as described in section 2.4.

## 2.3 Combining the LSTM with spatial convolution

Although the plain LSTM performs well in handling temporal sequences of point-like quantities it is not designed to recognize spatial features in sequences of two-dimensional maps as atmosphere-ocean interface fields. To address this limitation, we employ a ConvLSTM architecture as described below.

In this type of network, the elements of the input sequence are given as spatially varying fields $\boldsymbol{X}^t[\tau] = (X_k^t[x, y, \tau]) \in \mathbb{R}^{N_k \times (N_x \times N_y)}$, where $x \in [1, N_x]$ and $y \in [1, N_y]$ run over the horizontal and vertical dimensions of the map, respectively. To enable the "learning" of spatial patterns, the free parameters of the network are replaced by two-dimensional convolution kernels $\boldsymbol{\mathcal{M}}^g = (\mathcal{M}_{hk}^g[\xi, \eta]) \in \mathbb{R}^{(N_h \times N_k) \times (N_\xi \times N_\eta)}$ and $\boldsymbol{\mathcal{N}}^g = (\mathcal{N}_{hh'}^g[\xi, \eta]) \in \mathbb{R}^{(N_h \times N_h) \times (N_\xi \times N_\eta)}$. The width and the height of the kernels are given by $N_\xi$ and $N_\eta$, respectively and $\xi \in [-(N_\xi-1)/2, (N_\xi-1)/2], \eta \in [-(N_\eta-1)/2, (N_\eta-1)/2]$. Without loss of generality, we assume odd numbers for the kernel sizes.

A convolution with these kernels then gives the mapping from the input quantities to the gates.

$$g_h^t[x, y, \tau] = \mathcal{M}_{hk}^g[\xi, \eta] X_k^t[x-\xi, y-\eta, \tau] + \mathcal{N}_{hh'}^g[\xi, \eta] H_{h'}^t[x-\xi, y-\eta, \tau-1] + \mathcal{B}_h^g \tag{6}$$

again with Einstein's convention imposed.

It becomes immediately apparent that in case of the ConvLSTM, the gate and state vectors must become vector fields $(\in \mathbb{R}^{N_h \times (N_x \times N_y)})$ as well. We can write this mapping in the same way as (3), but by replacing the standard matrix-vector multiplication by a convolution (denoted with $*$), i.e.,

$$\boldsymbol{g}^t[\tau] = \boldsymbol{\mathcal{M}}^g * \boldsymbol{X}^t[\tau] + \boldsymbol{\mathcal{N}}^g * \boldsymbol{H}^t[\tau-1] + \boldsymbol{\mathcal{B}}^g \tag{7}$$

The subsequent processing of the $\boldsymbol{g}^t[\tau]$ remains symbolically the same as presented in (4) but with all appearing quantities now meaning vector fields instead of simple vectors.

In summary, the ConvLSTM is designed to process tasks that demand a combined understanding of spatial patterns and temporal sequences in data. It merges the image-processing capabilities of Convolutional Neural Networks (CNNs) with the time-series modeling of Long Short-Term Memory (LSTM) networks.

## 2.4 Fully connected layer

As stated in section 2.2.1, the final output $\boldsymbol{H}^t[N_\tau]$ and $\boldsymbol{C}^t[N_\tau]$ of the ConvLSTM encode information on the full input sequence. To contract this information to obtain the runoff vector $\boldsymbol{R}^t$ representing the $N_r$ rivers, we propose to subject the final short-term memory (i.e., the hidden state $\boldsymbol{H}^t[N_\tau]$) to an additional FC network.

In particular, the dimensionality of the vector field $\boldsymbol{H}^t[N_\tau]$ is sequentially reduced by three nested FC layers, each connected to the other by the Rectified Linear Unit (ReLU), see Figure 2. Integrating out artificial degrees of freedom in a step-wise fashion has turned out to be beneficial.

The runoff of the $r$-th river is then obtained via (using Einstein's convention)

$$R_r^t = \mathcal{W}_{rb}^3 \mathrm{ReLU}\left(\mathcal{W}_{ba}^2 \mathrm{ReLU}\left(\mathcal{W}_{ah}^1[x,y]H_h^t[x,y,N_\tau] + \mathcal{B}_a^1\right) + \mathcal{B}_b^2\right) + \mathcal{B}_r^3 , \tag{8}$$

where $a = 1, \ldots N_a, b = 1, \ldots N_b$ and the hyper parameters $N_a$ and $N_b$ are chosen such that $N_h \cdot N_x \cdot N_y > N_a > N_b > N_r$ in order to achieve the aforementioned step-by-step reduction of dimensionality. The weights $\mathcal{W}$ and biases $\mathcal{B}$ stand for parameters that are optimized during the training of the network.

In matrix-vector notation this can be compressed to,

$$\boldsymbol{R}^t = \boldsymbol{\mathcal{W}}^3 \mathrm{ReLU}\left(\boldsymbol{\mathcal{W}}^2 \mathrm{ReLU}\left(\boldsymbol{\mathcal{W}}^1 \boldsymbol{H}^t[N_\tau] + \boldsymbol{\mathcal{B}}^1\right) + \boldsymbol{\mathcal{B}}^2\right) + \boldsymbol{\mathcal{B}}^3 . \tag{9}$$

Combining equation (9) with equation (5) provides finally an explicit formula for the initial assumption of modeling the runoff for time $t$ as a functional of a sequence of atmospheric fields, i.e.

$$\begin{aligned}
\boldsymbol{R}^t &= \boldsymbol{M}(\{X_k^t[x,y,\tau]\}) \\
&= \boldsymbol{\mathcal{W}}^3 \mathrm{ReLU}\left(\boldsymbol{\mathcal{W}}^2 \mathrm{ReLU}\left(\boldsymbol{\mathcal{W}}^1 \boldsymbol{L}_H\left(\boldsymbol{X}^t[N_\tau], L\left(\boldsymbol{X}^t[N_\tau - 1], \ldots L\left(\boldsymbol{X}^t[1], (0,0)\right)\ldots\right)\right) + \boldsymbol{\mathcal{B}}^1\right) + \boldsymbol{\mathcal{B}}^2\right) + \boldsymbol{\mathcal{B}}^3 ,
\end{aligned} \tag{10}$$

where the $\boldsymbol{L}_H$ means that only the hidden state vector of the final ConvLSTM call is forwarded to the FC layer.

## 3 Technical details

### 3.1 Runoff data used for training

The non-stationary daily runoff data covering the period 1979 to 2011 is based on an E-HYPE hindcast simulation that was forced by a regional downscaling of ERA-Interim (Dee et al., 2011) with RCA3 (Samuelsson et al., 2011) and implemented into NEMO-Nordic (Hordoir et al., 2019) as a mass flux. The BMIP project (Gröger et al., 2022) played a crucial role in addressing the lack of consistent river discharge data for the entire study period (1961–2018). To this point, no comparable long-term dataset with daily resolution was available. In other studies multiple datasets have been merged, but offer only

monthly resolution (see e.g. Figure 3 (Meier et al., 2019)) . Hence, a new homogeneous runoff dataset was created. The 1961–1978 runoff data is based on Bergström and Carlsson (1994), with values interpolated from monthly to daily scales. The 2012-2018 data are derived from an E-HYPE forecast product. To ensure consistency for the analysis, the periods before (1961 to 1978) and after (2012 to 2018) have been neglected. Notably, the Neva River is an exception, as its discharge data originates from observational records (1961–2016) provided by the Russian State Hydrological Institute rather than E-HYPE hindcasts.

It should be noted that for this study, we used an intermediate dataset of river runoff developed during BMIP that was employed to run the ocean model. In this dataset, some rivers had not yet been merged, resulting in discrepancies between the number of freshwater input locations of 97 in this study and 91 rivers in the final version of Väli et al. (2019). The quality of the runoff was extensively evaluated. The dataset was found to closely align with historical observations for various rivers and with the Bergström and Carlsson (1994) dataset, showing a difference of under 1% for total Baltic Sea runoff (Väli et al.,

2019). For more information, see (Gröger et al., 2022) and (Väli et al., 2019).

### 3.2 Atmospheric Forcing

The UERRA-HARMONIE regional reanalysis dataset was developed as part of the FP7 UERRA project (Uncertainties in Ensembles of Regional Reanalyses, http://www.uerra.eu/). The UERRA-HARMONIE system represents a comprehensive, high-resolution reanalysis covering a wide range of essential climate variables. This data set includes data on air temperature,

pressure, humidity, wind speed and direction, cloud cover, precipitation, albedo, surface heat fluxes, and radiation fluxes from January 1961 to July 2019. With a horizontal resolution of 11 km and analyses conducted at 00 UTC, 06 UTC, 12 UTC, and 18 UTC, it also provides hourly resolution forecast model data. UERRA-HARMONIE is accessible through the Copernicus Climate Data Store (CDS, https://cds.climate.copernicus.eu/#!/home), initially produced during the UERRA project and later transitioned to the Copernicus Climate Change Service (C3S, https://climate.copernicus.eu/copernicus-regional-reanalysis-europe).

For the training of the neural network the hourly data was remapped to daily values.

Lastly, it should be noted that UERRA is not the atmospheric dataset that was used to drive the original E-HYPE model.

## 3.3 Ocean Model

We use use a coupled 3-dimensional ocean model, called the Modular Ocean Model (MOM) (Griffies, 2012) to simulate the Baltic Sea. It has a horizontal resolution of three nautical miles, roughly corresponding to 5.556 km and 152 vertical z* levels with a first layer thickness of 0.5m and a total depth of 264m. This model uses a finite-difference method to solve the full set of primitive equations to calculate the motion of water and the transport of heat and salt. The K-profile parametrization (KPP) was used as a turbulence closure scheme. The model's western boundary opens into the Skagerrak and connects the Baltic Sea to the North Sea. A more detailed description of the setup can be found in (Gröger et al., 2022).

## 3.4 Neural network hyper parameters

Our architecture is implemented as a sequential model, which allows for testing multiple ConvLSTM layers - a concatenation of multiple ConvLSTM cells. The following ConvLSTM output is then mapped by three fully connected linear layers, where the final layers map the output to 97 rivers. We used a custom loss function similar to a mean squared error loss that penalizes outliers stronger. For the training, we use the AdamW optimizer, which is an improved version of Adam that decouples weight decay (regularization) from the gradient update, likely leading to better generalization. The optimizer is configured with a learning rate and a weight decay to prevent overfitting. To adapt the learning rate during training, we employ the ReduceLROn-Plateau scheduler. This scheduler monitors the validation Mean Squared Error (MSE) and reduces the learning rate by a factor of 10 if no improvement is observed over 10 epochs. This dynamic adjustment helped the model converge more efficiently and avoid getting stuck in local minima.

The best set of hyper parameters have been defined by iterating over a pre-defined selection of possible parameters. The set of hyper parameters that has been chosen for the present study is given in Table 1. The model's performance can be described as relatively robust when changing the set of hyper parameters (see Figure C1). Interestingly, also shorter input sizes of 10 days perform really well. However, we still decided to use longer time scales, as we assume that longer input sizes increase the stability of the model needed for long-term climate simulations.

| Parameter name | Parameter size |
| --- | --- |
| Channel size | 4 |
| Num. hidden layer | 9 |
| Num. timesteps | 30 |
| Conv. kernel size | (7,7) |
| Num. ConvLSTM layers | 1 |
| Batch size | 50 |

**Table 1.** Final set of parameters for the ConvLSTM model

River runoff is influenced not only by the current day's atmospheric conditions but also by the cumulative and lagged effects of prior days' weather patterns. The choice of atmospheric fields was based on the assumption that the runoff should be

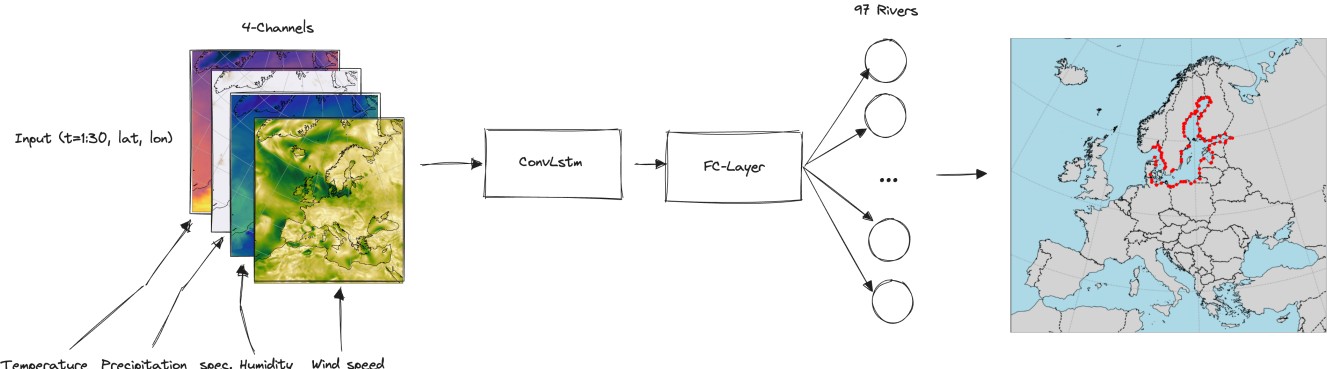

**Figure 4.** Schematic structure of the ConvLSTM implementation for river runoff forecasting.

mainly given by the net precipitation (precipitation minus evaporation). The evaporation flux is often calculated as a function of wind speed, the air's humidity, density and the involved turbulent exchange coefficients (Karsten et al., 2024), where the air temperature influences the latter two. Hence, as input, the model receives $N_\tau = 30$ days of atmospheric surface fields temperature, precipitation, specific humidity, and wind speed, with a daily resolution to predict daily river runoff $\boldsymbol{R}$ (see Figure 4). This window size allows the model to "remember" key atmospheric conditions leading up to a given day, enabling it to accurately predict runoff.

## 4 Results

### 4.1 ConvLSTM model evaluation

The model was trained and evaluated with daily data from 1979 to 2011, as this period represents the only period of E-HYPE without further bias correction applied to the runoff to match observations. The complete dataset was divided into randomly chosen splits of 80% training data, 10% validation data to evaluate the model's performance during training, and 10% test data which is finally used to asses the model's performance after training. The model was trained for 400 epochs, and the model weights with the lowest mean squared validation error have been stored. The model's accuracy for the combined daily predicted runoff from all 97 rivers flowing into the Baltic Sea is displayed in Figure 5. For this evaluation, the model's output is compared to the test data, which the model has not seen during the training phase.

The left panel a) illustrates the relative prediction error in relation to the E-HYPE data, indicating that, on daily timescales, the model can predict river runoff with an accuracy of $\pm 5\%$. The overall correlation is 0.997 with the resulting error metrics yielding a root mean square error (RMSE) of 323.99 $m^3$/s and mean absolute error (MAE) of 249.51 $m^3$/s. While the model's performance is already satisfactory, the discrepancies between the actual values and the predictions can partly be attributed to the use of a different atmospheric dataset than the one originally used to drive the E-HYPE model. However, by applying a rolling mean with a 5-day window, the prediction error is reduced to less than 1%, which is acceptable for climate modeling

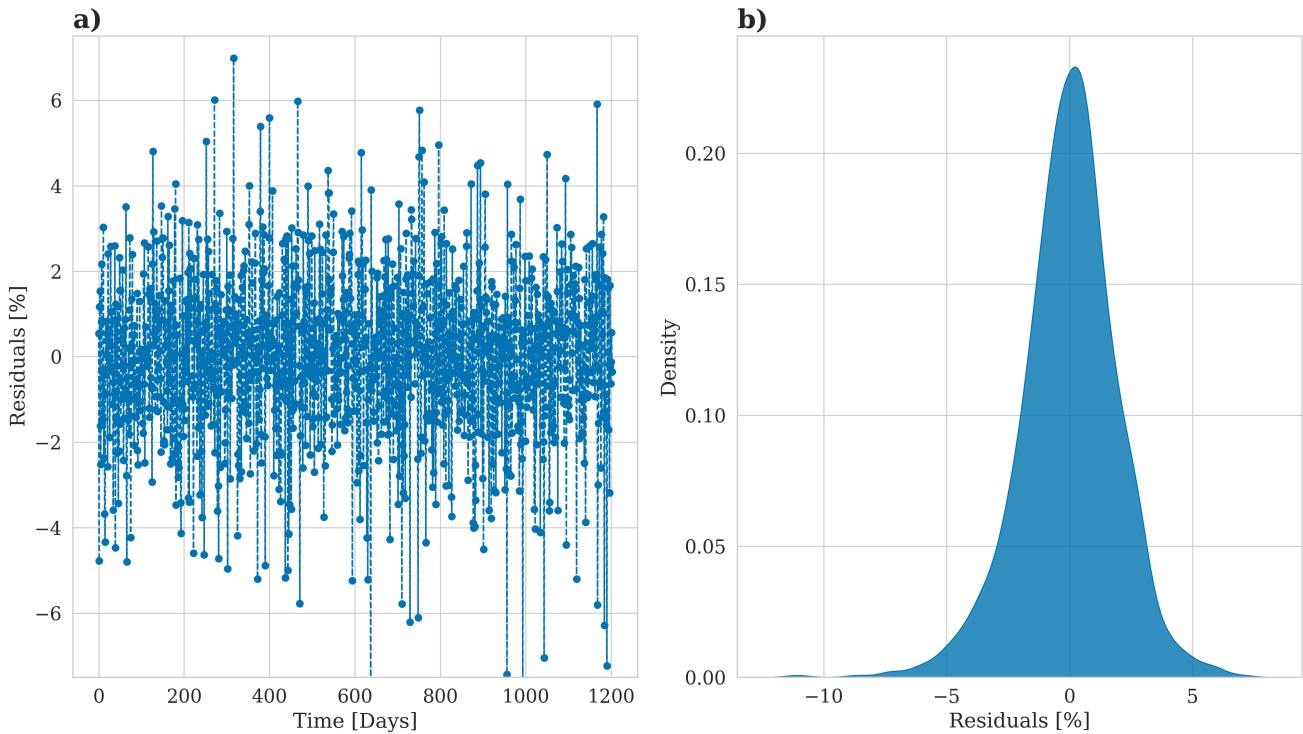

**Figure 5.** Model Accuracy for Predicted River Runoff. **(a)** Relative prediction error over time in relation to the E-HYPE data. **(b)** Density plot of residuals.

purposes. The right panel (b) displays the distribution of residuals as a density plot. The Figure shows that the distribution of residuals follows a Gaussian shape. The bell-shaped curve is approximately centered around zero indicates that the model does not exhibit a systematic bias, meaning it does not consistently overestimate or underestimate the river runoff values. Most residuals lie within a narrow range around zero, suggesting that large prediction errors are relatively rare.

For individual rivers, the distribution of the residuals still follows a Gaussian shape. However, on daily time scales the errors are larger reaching $\pm\,30\%$ during individual peaks D1.

In the following, the model's performance in reproducing the total river runoff and the discharges of four individual rivers into the Baltic Sea is addressed. Using the test dataset, Figure 6 shows the predicted river runoff and E-HYPE data.

Panel a) illustrates the total river runoff into the Baltic Sea, with both the predicted runoff (ConvLSTM) and the river runoff of the hydrological model E-HYPE, smoothed using a 5-day rolling mean. The predicted river runoff closely matches the original data, demonstrating the model's accuracy in predicting the overall river runoff into the Baltic Sea. Panel b) focuses on the Neva River (see 1), one of the largest rivers flowing into the Baltic Sea. The residual plot illustrates the prediction errors relative to the E-HYPE river runoff data over time. The ConvLSTM model predicts the Neva River runoff within a $\pm\%\,2.5\%$ range.

Panel c-e) shows the prediction residuals for the Oder River, the Umeälven River and the Neman River (see 1). Compared to the Neva River, the prediction errors are larger, which may be attributed to the training dataset, as the runoff for the Neva is based on measurements, whereas the other rivers are solely based on E-HYPE. Still, all results lie within the error margin of E-HYPE itself compared to observations (Supplementary Figure A1), with the average error on daily time scales for individual rivers mostly under 10%, showcasing the model's capability to forecast runoff for this river accurately.

The residuals were calculated as the relative difference between the predicted runoff and runoff data used for training and finally normalized by the runoff data used for training

An overview of the individual error metrics is give in Table C1.

## 4.2 Application of the ConvLSTM in combination with an ocean model

Lastly, we evaluate the performance of the ConvLSTM by incorporating the predicted river runoff as hydrological forcing into the ocean model MOM5. This provides a robust validation of the runoff model against more complex real world conditions and ensures that the predictions accurately reflect the impact of the river discharge on the ocean dynamics. This, in turn, validates that the ConvLSTM captures the temporal and spatial variability of river runoff and that the residuals shown in Figure 5 are indeed insignificant when it comes to realistic applications.

Figure 7 shows the salinity comparison between the E-HYPE and the predicted river runoff at BY15 - a central station in the Baltic Sea, east of Gotland island. The ocean model simulation using the predicted river runoff by the ConvLSTM closely mirrors the control simulation that is forced with the E-HYPE runoff. The upper panel (a) shows the surface salinity, representing the high-frequency variations in salinity heavily affected by river runoff. The predicted salinity using ConvLSTM river runoff matches the control simulation well, capturing the short-term fluctuations effectively. The lower panel (b) shows the bottom salinity, representing low-frequency variations in the Baltic Sea, which is also well reproduced with the ConvLSTM predictions. It should be noted that the discrepancies between the simulated salinity and the observed values at BY15 are not directly linked to the performance of the ConvLSTM river runoff model. Instead, it is attributed to the MOM5 ocean model's representation of physical processes, particularly the treatment of mixing, advection, and stratification in the Baltic Sea. Several factors may contribute to this discrepancy. The Baltic Sea is known for its strong vertical stratification due to the input of freshwater from rivers. The MOM5 model uses the K-profile parameterization (KPP) scheme for turbulence, which may not perfectly resolve small-scale mixing processes and vertical salinity gradients. This can result in an overestimation of salinity variability at the surface. Morever, while the MOM5 model captures the large-scale dynamics of the Baltic Sea, the lateral transport of saltwater from the Skagerrak into the central Baltic Sea may not be perfectly represented. This can introduce variability in surface and bottom salinity that is not observed in reality. However, all in all, the long-term trends and larger salinity changes are accurately captured, indicating the model's robustness in predicting high-frequency and low-frequency variations.

The final evaluation of the ConvLSTM model concentrates on the spatial accuracy of river runoff predictions as visualized in Figure 8. Panel a) exhibits the vertically averaged salinity from 1981 to 2011 in the reference simulation. It highlights the Baltic Sea's strong horizontal gradients and complex topographic features, as evidenced by salinity variations in deeper waters

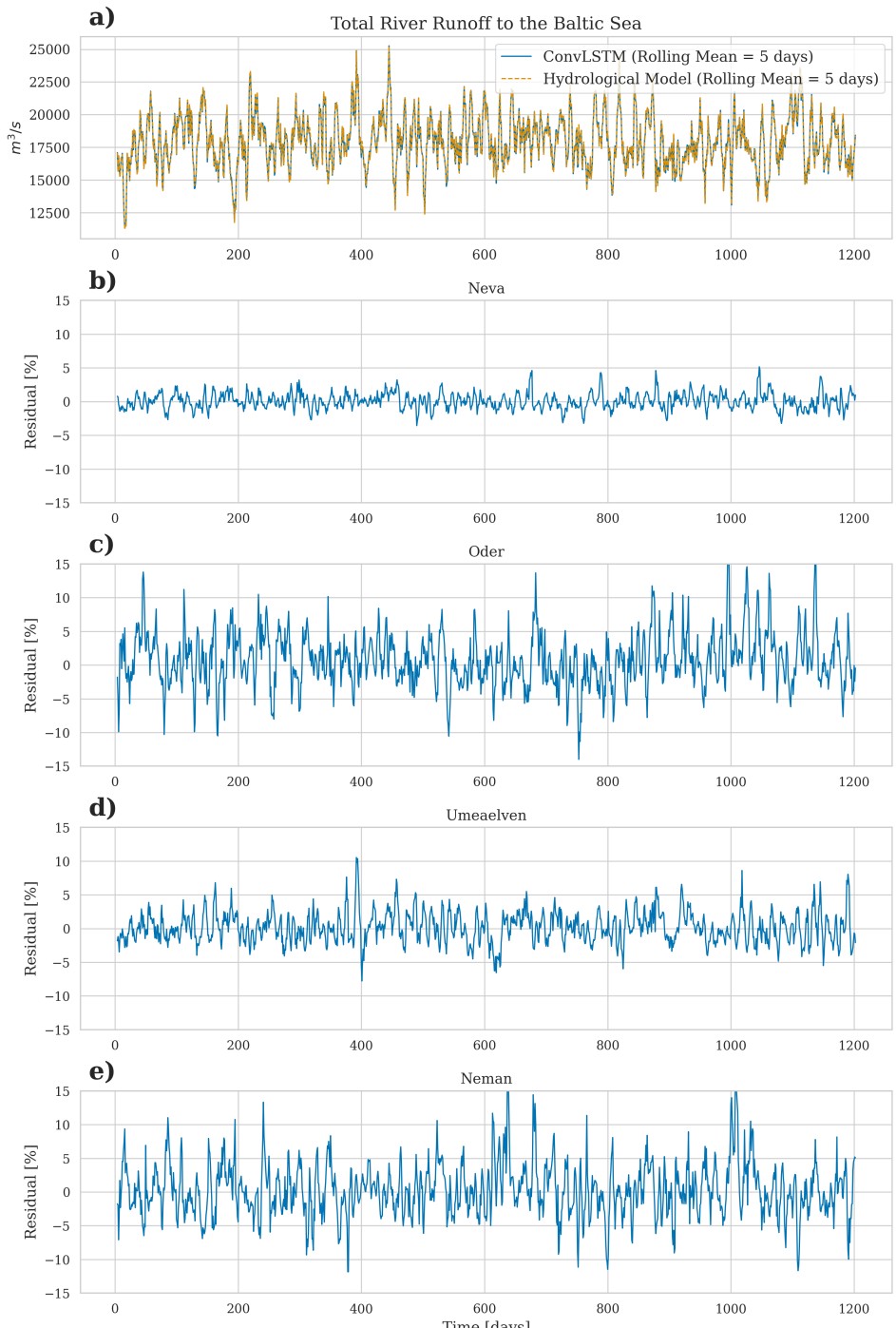

**Figure 6.** Model performance in predicting river runoff: **(a)** Total river runoff to the Baltic Sea with a 5-day rolling mean for both the predicted and original data of the hydrological model E-HYPE. **(b-e)** Residuals of runoff prediction for individual rivers showing the prediction error over time. The residuals were calculated as the relative difference between the predicted and observed values, normalized by the observed values.

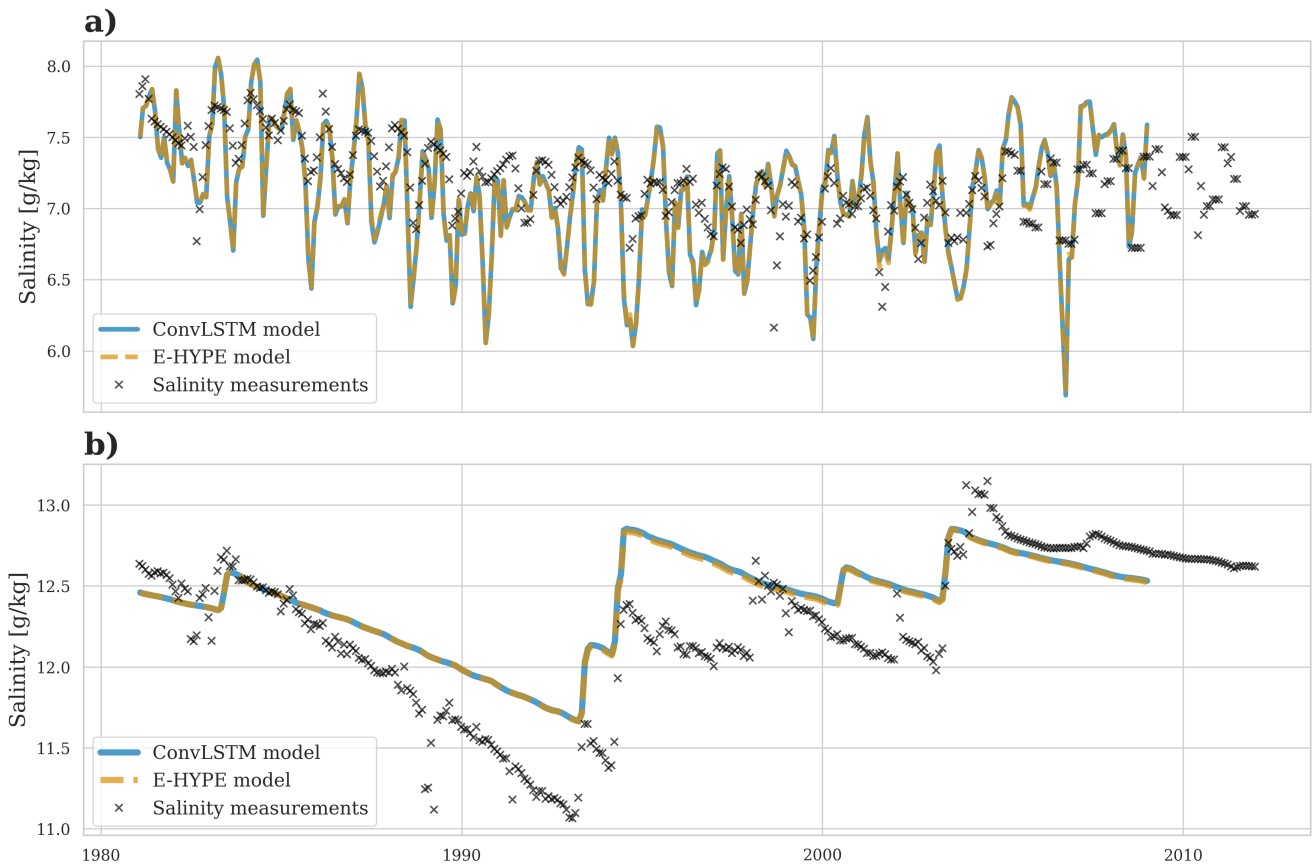

**Figure 7.** Salinity comparison at BY15 in the Baltic Sea: **(a)** Surface salinity showing high-frequency variations, and **(b)** Bottom salinity showing low-frequency trends. The ConvLSTM predictions closely follow the E-HYPE data, demonstrating the model's accuracy in reproducing salinity levels affected by river runoff.

captured by the vertical integration. In panel b), these reference results are compared to the ConvLSTM simulation by showing the percentage difference in vertically averaged salinity. Overall, the differences remain below 1%, except in the Gulf of Riga (22-24°E, 56.5-58.5°N for orientation), where the Daugava River dominates the runoff. The difference is approximately 1%.

## 5  Conclusions

With the increasing demand from decision makers for regional climate projections to quantify regional climate change impacts, the availability of precise hydrological forecasting becomes invaluable. In this work, we describe the implementation of a ConvLSTM network for predicting river runoff in a regional climate model, highlighting its potential to enhance river runoff

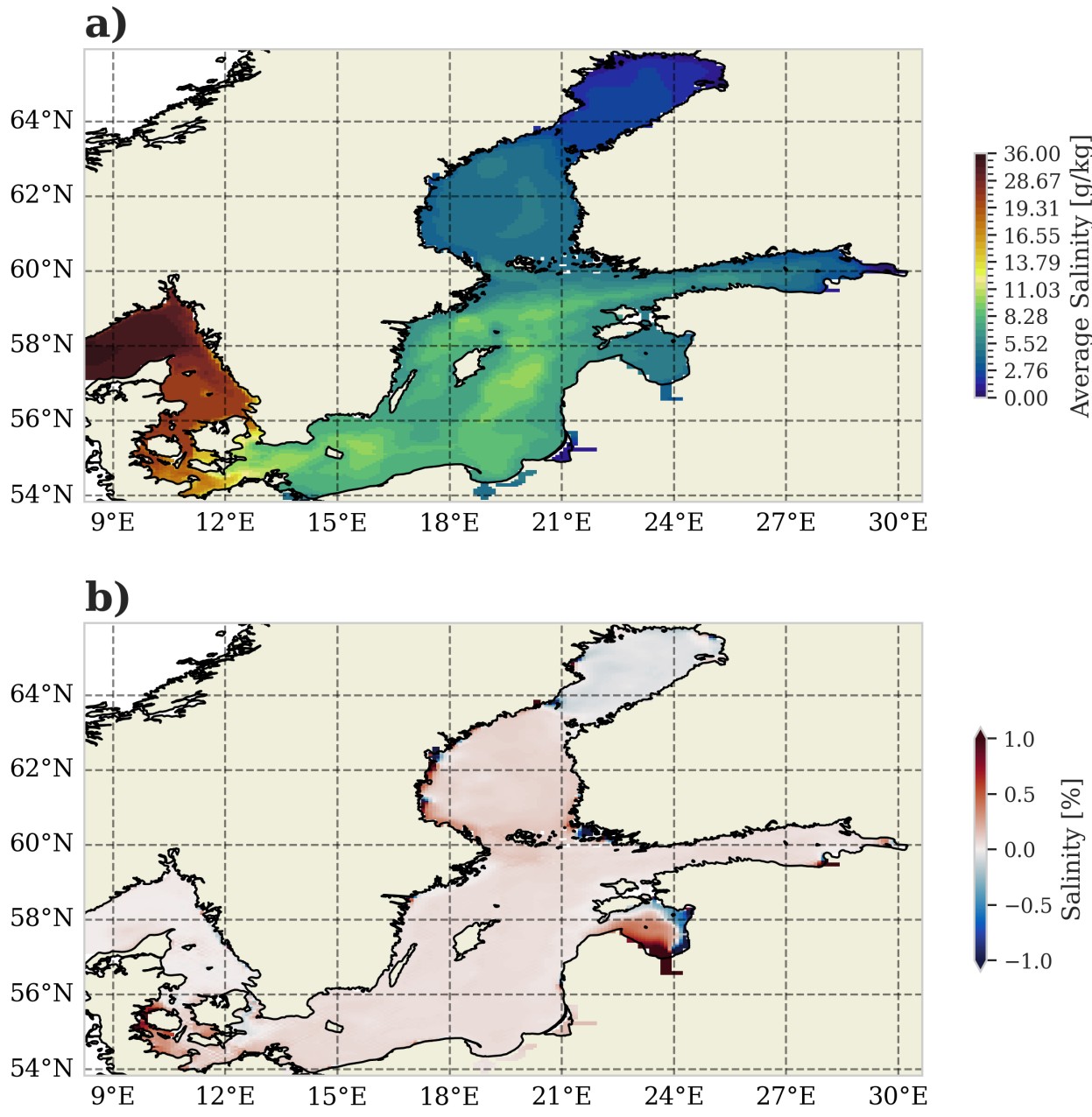

**Figure 8.** Spatial accuracy of river runoff predictions using the ConvLSTM model. **(a)** Vertically averaged salinity for the period 1981 to 2011 in the reference simulation, highlighting the strong horizontal gradients and complex topographic features in the Baltic Sea. **(b)** Percentage difference in vertically averaged salinity between the reference simulation and the ConvLSTM simulation.

forecasting across different coastal seas. Our model not only reproduces the total river runoff entering the Baltic Sea but also performs exceptionally well for individual rivers.

All results lie within the error margin of the hydrological model itself when compared to observations, with the average error on daily time scales for individual rivers mostly under 10%. Hence, our results confirm the excellent performance of LSTM networks in predicting river runoff (Humphrey et al., 2016; Huang et al., 2014; Ashrafi et al., 2017; Liu et al., 2020; Fang and Shao, 2022; Kratzert et al., 2018). In addition, our results align well with the observed performance of ConvLSTMs in similar applications for predicting single rivers (Ha et al., 2021), basin-wide runoff (Zhu et al., 2023), and precipitation now-casting (Shi et al., 2015). In our study, we further extend the use of ConvLSTM's by predicting multiple (n=97) rivers at once, while maintaining high accuracy for the entire Baltic Sea as well as for individual rivers. Moreover, the predicted river runoff proved robust when using the river runoff in an comprehensive ocean model setup of the Baltic Sea. Extending the simulation beyond the known period also provided robust results (Supplementary Figure B2).

The transition from traditional hydrological models to machine learning approaches, such as the implement ConvLSTM model, offers significant advantages as the model can be seamlessly integrated into regional climate models, allowing for real-time computation of river runoff while performing climate projections. While the initial training of the model requires substantial computational resources, it remains significantly less intensive than running comprehensive hydrological models. Furthermore, once trained, the ConvLSTM model is computationally efficient during inference, ensuring enhanced forecasting capabilities without significantly increasing computational demands. The achieved speedup (depending on the complexity of the hydrological model) is within the range of 30 to 90 times faster.

Nevertheless, the quality of the ConvLSTM model still depends on the performance of the hydrological model, which provides a comprehensive, homogeneous dataset essential for effective training. While different in their architecture, both the hydrological model and the ConvLSTM model provide a mapping from atmospheric variables to river runoff. This training approach contrasts with using measurement data for training, which is significantly more challenging due to the data sparsity over larger regions and potentially varying measurement techniques. Thus, rather than rendering traditional methods obsolete, the integration of machine learning models builds upon and enhances the foundational data provided by hydrological models methods.

The robust performance of the ConvLSTM model in simulating river runoff and its possible effective integration into coupled regional climate models, as for example in the IOW ESM (Karsten et al., 2024), paves the way for a multitude of new storyline simulations. Importantly, this can now be achieved without any expert's domain knowledge on hydrological modelling. Hence, we can now explore various "what-if" scenarios more reliably, under the assumption that the model weights attained during training are robust. Such scenario testing is crucial for crafting effective water resource management strategies and adapting to a changing climate and hence represents a significant step forward in our ability to understand and predict the complex dynamics of river systems and their impact on regional climate systems.

*Code and data availability.* The ocean model data and atmospheric data used in this study is described and/or published in (Gröger et al.,
2022). All additional data is properly cited and accessible. The ocean model data needed to reproduce the results of this study is accessible
at https://zenodo.org/records/13365099. The code of the ConvLSTM can be accessed at https://zenodo.org/records/13910136 All scripts
to reproduce the figures can be found at https://zenodo.org/records/13910136 The source code of the ocean model is available at https:
//github.com/mom-ocean/MOM5

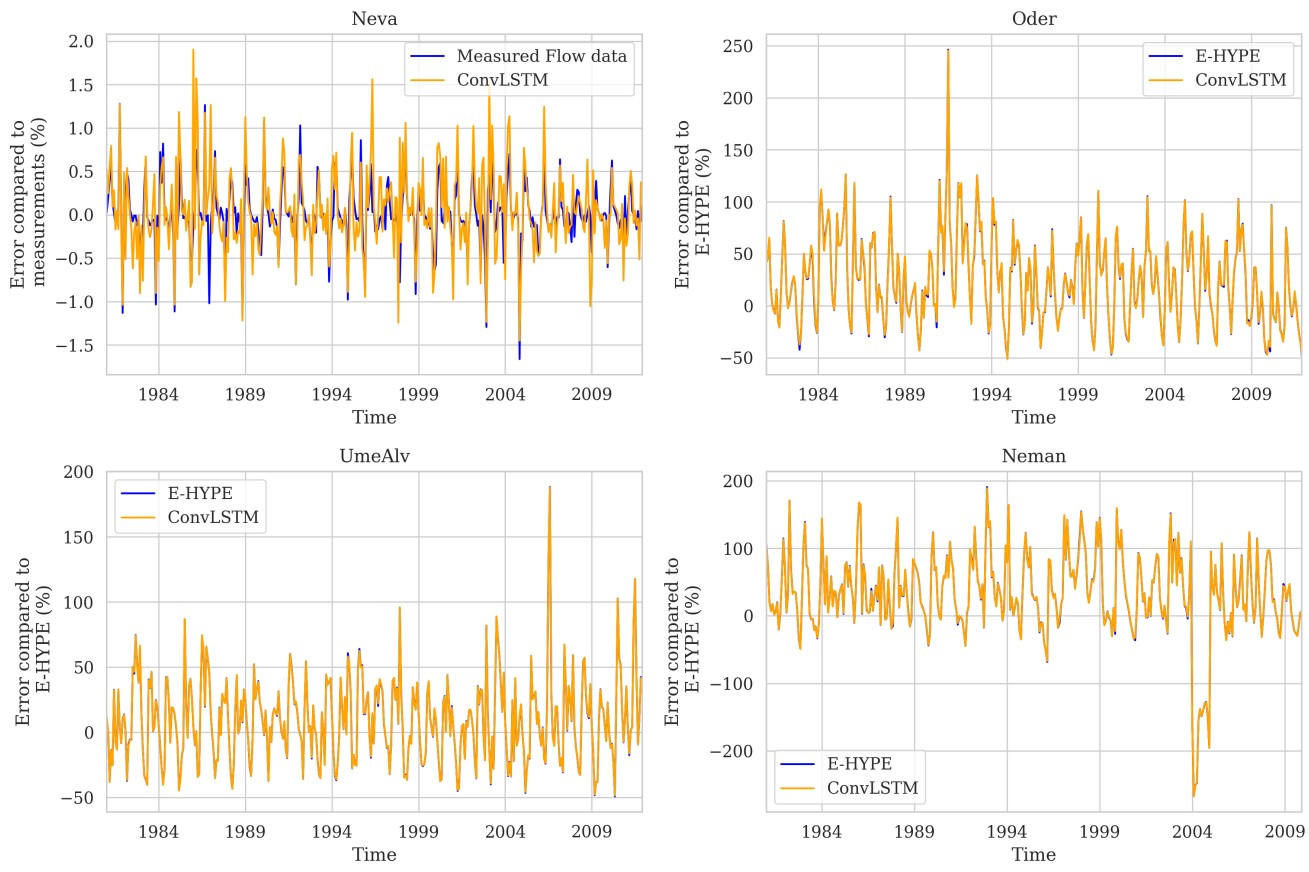

**Figure A1.** Residual for the hydrological model E-HYPE as well as for the prediction by the ConvLSTM model. The residuals were calculated as the relative difference between the predicted and observed values, normalized by the observed values.

## Appendix A:  Appendix A

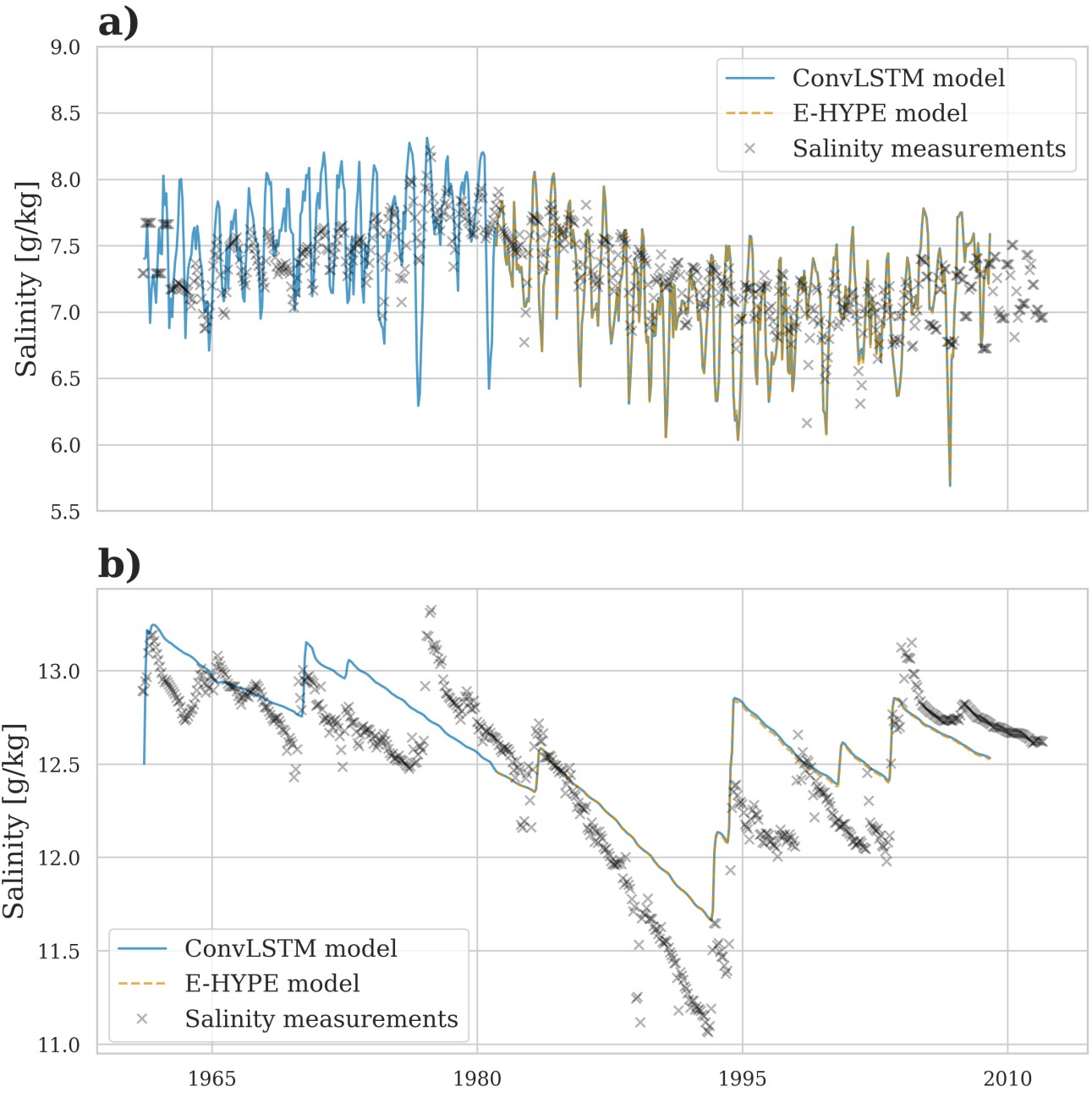

**Figure B1.** Extended regional ocean model simulation for the period 1961-2009. The station BY15 is validated.

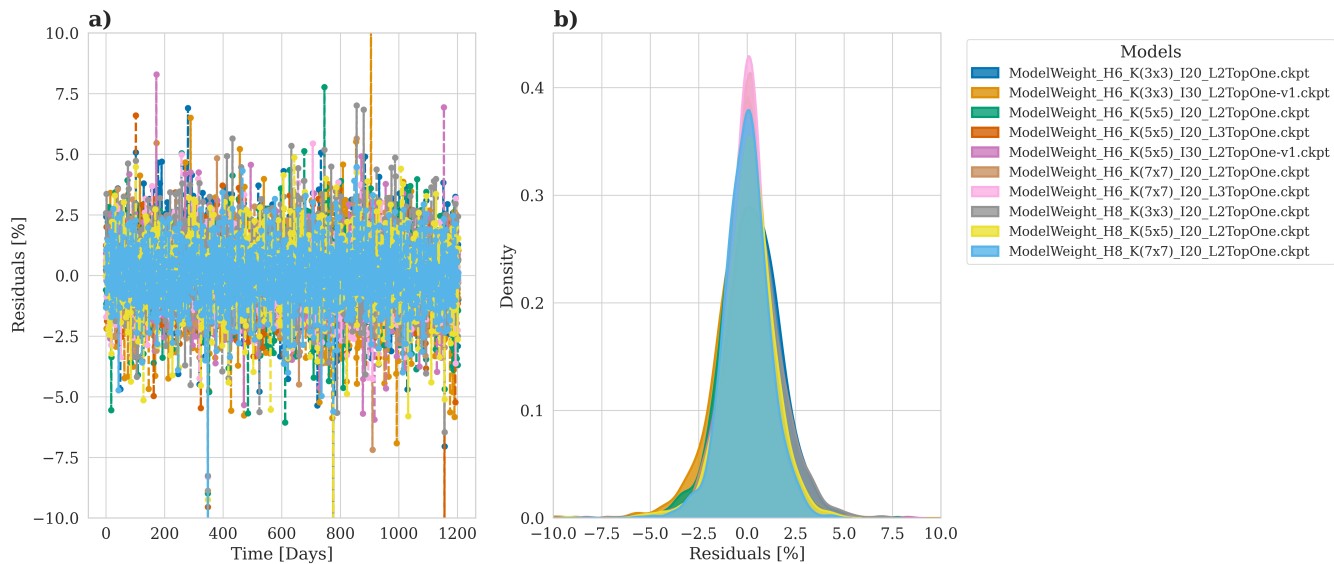

**Figure C1.** Model Accuracy for Predicted River Runoff for 10 different hyper parameters. (a) Relative prediction error over time in relation to the E-HYPE data. (b) Density plot of residuals

| River | RMSE $m^3$/s | MAE $m^3$/s | Correlation |
|---|---|---|---|
| All Rivers | 323.99 | 249.51 | 0.997 |
| Neva | 65.41 | 48.36 | 0.995 |
| Oder | 49.93 | 38.79 | 0.994 |
| Umaelven | 23.73 | 17.92 | 0.994 |
| Neman | 55.65 | 41.63 | 0.996 |

**Table C1.** Model performance in predicting river runoff, error metrics

*Author contributions.* FB: Writing – review & editing, Writing – original draft, Visualization, Resources, Methodology, Investigation, Formal analysis, Conceptualization. SK: Writing – review & editing, Methodology. KR: Visualization, Writing – review & editing, Methodology. UG: Writing – review & editing, Conceptualization.

*Competing interests.* The authors declare no competing interest.

*Disclaimer.* TEXT

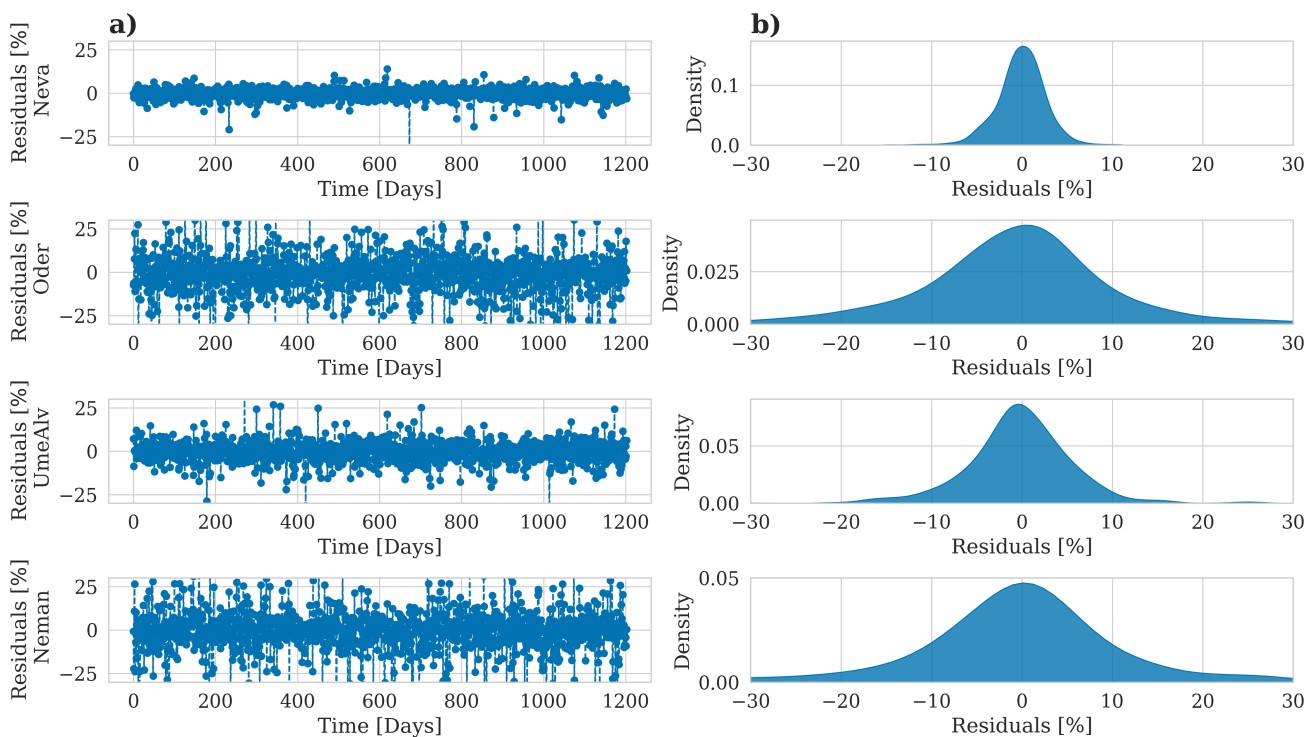

**Figure D1.** Model Accuracy for Predicted River Runoff for individual rivers. (a) Relative prediction error over time in relation to the E-HYPE data. (b) Density plot of residuals

*Acknowledgements.* The research presented in this study is part of the Baltic Earth program (Earth System Science for the Baltic Sea region, see [https://www.baltic.earth](https://www.baltic.earth/). The authors gratefully acknowledge the computing time granted by the Resource Allocation Board and provided on the supercomputers at NHR@ZIB as part of the NHR infrastructure.

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
