# Peer review of "From Weather Data to River Runoff: Using Spatiotemporal Convolutional Networks for Discharge Forecasting"

_EGUsphere, 2024_

## Author Comment (AC1)

**Reviewer comments – Reply – Changes to the manuscript**

**Reviewer #1:** Journal: Geophysical Model Development (GMD)

Please note that all following line references correspond to the article without track changes.

General Comments: The authors present a machine learning approach to forecasting river runoff from weather data using convolutional long-short-term memory neural networks. They present convincing evidence that the utilized ml model shows results of equal quality as its training data. At the same time the ml method offers faster processing speeds and thus an easier direct integration into regional climate models. With this approach, they present a scientifically significant and qualitative contribution to the integration of river runoff forecasting into climate models. While the manuscript shows great potential, I think that it requires minor revisions. My comments are listed below.

Minor Comments: As I do not possess a deeper understanding of the river runoff modelling and come from the machine learning side, I will limit my comments mostly to the technical aspects.

First, to me it does not become clear exactly how well your training data performs in comparison to other state-of-the-art models. I understand that your ConvLSTM is able to reproduce its training data's quality but I'm not fully able to grasp the strengths and weaknesses of the utilized training model, which I can assumed are transferred to the ML model. It would be helpful to extend the technical details section or the model section by a short description of the training data and especially its strengths and weaknesses compared to other possible runoff forecasting models. Although I see that the point of the paper is more the proof that is able to reproduce a state-of-the-art river runoff forecasting and not the exact strenghts and weaknesses of the utilized training data, it would help give perspective to the strengths of your method

We agree that this is a good idea and added the following text to the paper.

L168-170: To this point, no comparable long-term dataset with daily resolution was available. In other studies multiple datasets have been merged, but offer only monthly resolution (see e.g. Figure 3 \citep{Meier2019}) .

For example, your training data seems to present a bias compared to observational data (Figure 7b), which the network reproduces.

You are right for pointing this out. However, this bias is likely caused by other factors than the river runoff. We added the following part for clarification:

L271-281: It should be noted that the discrepancies between the simulated salinity and the observed values at BY15 are not directly linked to the performance of the ConvLSTM river runoff model. Instead, it is attributed to the MOM5 ocean model's representation of physical processes, particularly the treatment of mixing, advection, and stratification in the Baltic Sea.

Several factors may contribute to this discrepancy. The Baltic Sea is known for its strong vertical stratification due to the input of freshwater from rivers. The MOM5 model uses the K-profile parameterization (KPP) scheme for turbulence, which may not perfectly resolve small-scale mixing processes and vertical salinity gradients. This can result in an overestimation of salinity variability at the surface. Moreover, while the MOM5 model captures the large-scale dynamics of the Baltic Sea, the lateral transport of saltwater from the Skagerrak into the central Baltic Sea may not be perfectly represented. This can introduce variability in surface and bottom salinity that are not observed in reality. However, all in all, the long-term trends and larger salinity changes are accurately captured, indicating the model's robustness in predicting high-frequency and low-frequency variations.

In connection to that, you describe that you utilize the time period from 1979 to 2011, because they are not bias corrected. As a bias correction seems to be usually conducted, I would like to know if that can be similarly performed on the ConvLSTM outputs.

Yes, in principle this bias correction is possible, likely by additional scaling as post-processing. On the other hand, some of these bias corrections have been performed due to reasons like changes to the water-management or a new dam. This is why we chose to use a period where none of the bias corrections have been applied.

Connected to that, have you tried to train the ConvLSTM on any other runoff models? Training for 400 epochs on daily training data from 32 years is a lot of training input. Just out of interest, have you tried training on less data and how does the performance of the ConvLSTM differ? I would guess, that not all hydrological models provide such a comprehensive dataset. Could you thus comment on how easy it would be to extend this method to other runoff prediction models and how much training data would be required.

We did some preliminary test with other measurement data, but the results we mixed. Rivers that have a good temporal coverage with no changes in the way the data was measured performed well, while other rivers where structural changes were done performed poorly. Hence, we assume that the good performance of the ConvLSTM is also based on the good data availability.

For this review we also used only 10 years of training data (E-HYPE). In its current configuration the model's performance is worse when only 1/3 of the trainings data is used.

[Figure]

[Figure]

I would also be interested, if all ocean/regional climate models are able to utilize runoff predictions from similar sources or if they require their own in-model consistent runoff forcing. Because, if other climate models would require the ConvLSTM to be trained on different runoff predictions, it would significantly limit this method's applicability if that runoff model would be required to possess such a comprehensive training dataset as the EHYPE model presented in your study.

In general, the runoff data (all individual rivers) are mapped onto the ocean grid as a mass flux. While the grids may differ, the procedure is similar across all major ocean models.

Additionally, I would be interested out of curiosity how many timesteps are necessary for the LSTM to significantly improve the CNN output. Have you tried training with significantly less than 30 timesteps? What was your reasoning behind choosing these 30 days? Or was it just based on model performance/loss functions?

Based on your suggestion we performed several sensitivity tests of the hyper parameters. The model performance turns out to be relatively robust, even for shorter time steps (10 days). However, we still decided to use longer time scales, as we assume that longer input sizes increase the stability of the model needed for long-term climate simulations.

L209-213 The model's performance can be described as relatively robust when changing the set of hyper parameters (see Figure \ref{fig-Supp3}. Interestingly, also shorter input sizes of 10 days perform really well. However, we still decided to use longer time scales, as we assume that longer input sizes increase the stability of the model needed for long-term climate simulations.

Finally, you claim that "While the initial training of the model requires substantial computational resources, it remains significantly less intensive than running comprehensive hydrological models" (Page 17). Could you give an estimate on how big

this"significant" reduction of computational resources is? Because in the end this time saving is the important improvement of your method compared to other numerical prediction systems/models.

We agree that this information is useful and added it to the text. Our model generates one year of daily river runoff in roughly 10 seconds. The runtime of a hydrological model (personal communication with Stefan Hagemann (Dr. Stefan Hagemann, Regional Land and Atmosphere Modeling, Head of Department), with a similar resolution varies between 5-15 minutes per year.

This results in a speed up in the range of factor 30-90.

L309-310: The achieved speedup (depending on the complexity of the hydrological model) is within the range of 30 to 90 times faster.

In general I felt the content of the paper was novel and the method would be of interest to others in the field, but some details should be explained further or lack a bit of background information.

Thank you very much and also thank you for taking the time to review the article.

---

## Author Comment (AC2)

**Reviewer comments – Reply – Changes to the manuscript**

**Reviewer #2:** Journal: Geophysical Model Development (GMD)

Please note that all following line references correspond to the article without track changes.

**General Comments:**
The authors introduce an impressive new way to forecast river runoff using ConvLSTM network models at the scale of the Baltic Sea catchment. Their study demonstrates that ConvLSTM can accurately predict daily runoff for 97 rivers in the Baltic Sea area by using weather data. The study shows the trained ConvLSTM model, on daily timescales, can predict river runoff with an accuracy of ± 5% compared to the original E-HYPE data. Impressively, the ConvLSTM model performs just as well as traditional hydrological models in capturing runoff patterns, but with faster processing and greater computational efficiency. This makes it a great fit for integration into regional climate models, enabling real-time runoff forecasting and improving the accuracy of coastal climate impact predictions. Overall, the authors make a strong case for ConvLSTM networks being well suited for integrating in regional climate models and a valuable tool for real-time river runoff prediction during climate projections. The ConvLSTM model proves reliable when using the river runoff in a comprehensive ocean model of the Baltic Sea to predict salinity.
While I think this is a strong article, I have a few minor comments that I think could enhance it.

**Specific Comments:**
My suggestions for the technical sections are as follows:

- In Väli et al. (2019), they originally generated 97 potential freshwater input locations from rivers in the Baltic Sea area, but this was later reduced to 91 in the final dataset. Could you clarify this discrepancy and explain why you state that 97 inputs are used in the ConvLSTM model?

  You are right. Thank you for spotting this error. We used an intermediate dataset of river runoff that was created during the BMIP project and was used to run the ocean model. In this dataset, some of the rivers have not yet been merged. We added this information to the manuscript.

  L175-180: It should be noted that for this study, we used an intermediate dataset of river runoff developed during BMIP that was employed to run the ocean model. In this dataset, some rivers had not yet been merged, resulting in discrepancies between the number of freshwater input locations of 97 in this study and 91 rivers in the final version of \citet{vali2019river}. The quality of the runoff was extensively evaluated. The dataset was found to closely align with historical observations for various rivers and with the \citet{Bergstrom897362} dataset, showing a difference of under 1\% for total Baltic Sea runoff \citep{vali2019river}. For more information, see \citep{groger2022} and \citep{vali2019river}.

- In the "Runoff Data for Training" section, it would be useful to add context around why the BMIP project runoff data is necessary. Specifically, it would help to mention

that a new homogeneous runoff dataset was created because no consistent river discharge data was available for the full period (1961–2018), and the E-HYPE model originally only covered a few recent years.

We agree and added this information to the text.

L168-170: To this point, no comparable long-term dataset with daily resolution was available. In other studies multiple datasets have been merged, but offer only monthly resolution (see e.g. Figure 3 \citep{Meier2019}).

- In the "Runoff Data for Training" section, while you clearly state that the 1979-2011 period of E-HYPE hindcast simulation data is used, it would improve clarity to specify that this data is on a daily scale. Additionally, while you indicate which periods were excluded (1961-1978 and 2012-2018), providing more insight into the spatial and temporal adjustments applied to these excluded periods would help to justify the selection the 1979-2011 data. For example:
  - The 1961-1978 data, based on Bergstrom & Carlsson (1994), was interpolated from monthly to daily values.
  - The 2012-2018 data is an E-HYPE forecast product, but further clarification on why this recent data was omitted would be beneficial.
  - Additionally, as noted in Väli et al. (2019), the Neva River is an exception, with its data coming from observational records (1961-2016) from the Russian State Hydrological Institute, rather than E-HYPE hindcasts. This exception should be explicitly highlighted as it is one of the four river locations evaluated in detail.
  - Finally, In the "Runoff Data for Training" section, the statement that the "quality of the runoff was extensively evaluated" is a bit broad. Since you are comparing ConvLSTM model output against E-HYPE hindcast data, it would help to detail the methods used for this evaluation. Including a statement about confidence in the BMIP data would also be valuable. Specifically, you might note from Väli et al. (2019) that the BMIP dataset closely aligns with historical observations for various rivers and with Bergstrom & Carlsson (1994)'s dataset, showing a difference of under 1% for total Baltic Sea runoff. This would reinforce the reliability of the BMIP data in the ConvLSTM modelling.

Thank you so much for these insights, which we added to the document. This helped clarify this section.

There are several changes from L165-180:

The non-stationary daily runoff data covering the period 1979 to 2011 is based on an E-HYPE hindcast simulation that was forced by a regional downscaling of ERA-Interim \citep{dee2011era} with RCA3 \citep{samuelssonRossbyCentreRegional2011} and implemented into NEMO-Nordic \citep{hordoir2019nemo} as a mass flux. The BMIP project \citep{groger2022} played a crucial role in addressing the lack of consistent river discharge data for the entire study period (1961–2018). To this point, no comparable long-term dataset with daily resolution was available. In other studies multiple datasets

have been merged, but offer only monthly resolution (see e.g. Figure 3 \citep{Meier2019}). Hence, a new homogeneous runoff dataset was created. The 1961–1978 runoff data is based on \citet{Bergstrom897362}, with values interpolated from monthly to daily scales. The 2012-2018 data are derived from an E-HYPE forecast product. To ensure consistency for the analysis, the periods before (1961 to 1978) and after (2012 to 2018) have been neglected. Notably, the Neva River is an exception, as its discharge data originates from observational records (1961–2016) provided by the Russian State Hydrological Institute rather than E-HYPE hindcasts.

It should be noted that for this study, we used an intermediate dataset of river runoff developed during BMIP that was employed to run the ocean model. In this dataset, some rivers had not yet been merged, resulting in discrepancies between the number of freshwater input locations of 97 in this study and 91 rivers in the final version of \citet{vali2019river}. The quality of the runoff was extensively evaluated. The dataset was found to closely align with historical observations for various rivers and with the \citet{Bergstrom897362} dataset, showing a difference of under 1\% for total Baltic Sea runoff \citep{vali2019river}. For more information, see \citep{groger2022} and \citep{vali2019river}.

- In the "Atmospheric Forcing" and "Ocean" sections confirm the temporal resolution of the Essential Climate Variables (ECV) are daily.

  We added this information to the text.

- In the "Atmospheric Forcing" and "Ocean" sections, the horizontal resolution of the models I think should be expressed in consistent units.

  We added this information to the text.
  L190: For the training of the neural network the hourly data was remapped to daily values.

- It would be clearer to use kilometres (km) throughout rather than miles.

  Added to the text.
  L194-195: It has a horizontal resolution of three nautical miles, roughly corresponding to 5.556 km and 152 vertical z* levels with a first layer thickness of 0.5m and a total depth of 264m.

- The ConvLSTM model was trained and tested using daily data from 1979 to 2011, with 80% for training, 10% for validation, and 10% for testing. It performed well on both the training and test data. Have you thought about how reducing the training data might impact the model's performance? This could give you some insight into the model robustness with less data.

  We tested only 10 years of training data (E-HYPE). In its current configuration the model's performance is worse when only 1/3 of the trainings data is used. It is beyond the scope of the current study but in practice, we might try to reduce the complexity

of the model and train for longer periods. We attached the corresponding figures below.

[Figure]

[Figure]

- It might be helpful to mention that the rivers feeding freshwater into the Baltic Sea have runoff data that is not stationary. One of the benefits of using LSTM models over other machine learning (ML) methods is that they're specifically designed to capture patterns and dependencies in sequences, making them a great fit for non-stationary data like this.
  Thank you for the hint. Mentioned now.
  L165-167: The non-stationary daily runoff data covering the period 1979 to 2011 is based on an E-HYPE hindcast simulation that was forced by a regional downscaling of ERA-Interim \citep{dee2011era} with RCA3 \citep{samuelssonRossbyCentreRegional2011} and implemented into NEMO-Nordic \citep{hordoir2019nemo}.

- Could you comment on alternative ML models that might be suitable for runoff prediction for freshwater inputs into the Baltic Catchment.

  For our research we assumed that we need a model that can capture spatio-temporal features, this is why we use the ConvLSTM structure. In principle, it should be possible to use a regular LSTM architecture. However, capturing relevant spatial-temporal features is then not guaranteed as you need to flatten the spatial features into a one-dimensional vector. Depending on the quality of the data, the temporal coverage, and the complexity of the research question simpler fully connected networks are likely also possible to predict single rivers.
  For example, preliminary test for single rivers with good temporal data coverage (measurements of the Warnow river near Warnemünde at the German coastline), indicates that LSTMs may even outperform E-HYPE. A single grid point of ERA5 (atmospheric data) over the Warnow region was used for the training.

[Figure]

Figure 6: River discharge prediction of the LSTM network for the evaluation data. The shaded gray area indicates one standard deviation of the test data. The overall correlation amounts to 0.9 of the LSTM and the original data. E-hype has an overall correlation of 0.83.

- The paper goes into a lot of detail about the ConvLSTM model architecture in the 'Implemented model architecture' section (Section 2), but the 'Neural network hyperparameters' section 3.4 could use a bit more explanation. It would be helpful to explain the model architecture a bit more such as whether a sequential model was used, which lets you stack layers in a simple, linear way. Also, it would help to clarify how the hyperparameter values in Table 1 were determined.

  The core of the model is a ConvLSTM cell that processes the input atmospheric data, which is provided as a sequence of spatial maps over time. The ConvLSTM architecture uses a convolutional operation to extract spatial features from the input while leveraging the LSTM's recurrent mechanism to capture temporal relationships. The final hidden state of the ConvLSTM captures the spatiotemporal features relevant to runoff prediction.

  After the ConvLSTM block, the output is flattened and passed through three fully connected (dense) layers. These layers reduce the dimensionality and map the learned spatiotemporal features to the target output: daily runoff values for 97 rivers.

  The architecture is implemented as a sequential model.

  L200-202: Our architecture is implemented as a sequential model, which allows for testing multiple convLSTM layers - a concatenation of multiple convLSTM cells. The best set of hyper parameters have been defined by iterating over a pre-defined selection of possible parameters.

- Neural network hyperparameters section 3.4, Table 1 shows details for only one layer, but it is unclear how many units (neurons) were in that layer. Did you consider adding more layers to help the model capture more complex patterns? Also, did you include

a Dropout layer after the LSTM layer to help prevent overfitting by randomly dropping some neurons during training? For the Fully Connected Layer, did you use a dense layer to create the final output? And when compiling the ConvLSTM model, what loss function (like MSE or MAE) and optimizer (e.g., Adam) did you use?

Determining the number of units (neurons) in a layer is not straightforward. In the case of applying a ConvLSTM, we utilize two-dimensional convolution kernels, denoted as $M_g$ and $N_g$ in the manuscript, for a given gate $g$ in the LSTM structure. The sizes of these kernels are determined by the number of input channels $k$ and the specified number of hidden dimensions $h$, as described in Equation 6 of the manuscript. Subsequently, the input undergoes a mapping in a fully connected layer, reducing dimensionality from the output of the last hidden state to 512 neurons to 256 neurons, to the final output, which represents 97 rivers.

We considered adding more layers (see the answers below to hyper parameter testing). One short coming was that the network structure during training grew too large for the GPU, if many hidden layers are considered. We tested two and three layers with smaller input dimensions (timesteps to consider), hidden dimensions, and kernel sizes. The results were also promising, and we think going forward with a slightly different model architecture would also be possible (more layers, smaller hidden dimensions). The more complex the model was designed the longer it took to converge during training. Especially, when more layers were considered. We decided to go for the best results that were to our experience strongly influenced by the kernel size.

We did consider using Dropout, however, overfitting was not received as a large problem. We did use weight decay in the Adam Optimizer during training.

We used a denser layer to create the final output:

```
1. self.river_predictors = nn.Sequential(
2.         nn.Linear(self.linear_dim, 512),
3.         nn.ReLU(),
4.         nn.Linear(512, 256),
5.         nn.ReLU(),
6.         nn.Linear(256, 97)
7.         )
8.
9.
```

For the loss we used a custom MSE loss that penalizes larger errors more heavily.

```
1. class EnhancedMSELoss(nn.Module): def __init__(self, alpha=1.5):
2. """ Initialize the enhanced MSE loss module. Args: alpha (float): Exponential factor to increase penalty for larger errors. """
3. super(EnhancedMSELoss, self).__init__() self.alpha = alpha
4. def forward(self, predictions, targets):
5. """ Calculate the enhanced MSE loss. Args: predictions (torch.Tensor): The predicted values. targets (torch.Tensor): The ground truth values. Returns: torch.Tensor: The calculated loss. """
6. error = predictions – targets
7. mse_loss = torch.mean(error**2)
8. enhanced_error = torch.mean(torch.abs(error) ** self.alpha)
```

The optimizer:

```
1. def configure_optimizers(self):
```

```
2.          """
3.          Configures the optimizer and learning rate scheduler.
4.
5.          Returns:
6.              tuple: List of optimizers and list of learning rate schedulers.
7.          """
8.          opt = torch.optim.AdamW(self.parameters(), lr=self.learning_rate, weight_decay=1e-4)
9.          sch = torch.optim.lr_scheduler.ReduceLROnPlateau(opt, mode='min', factor=0.1,
patience=10, verbose=False)
10.          return {"optimizer": opt, "lr_scheduler": sch, "monitor": "val_mse"}
11.
```

We also added more information to the text in section 3.4.

- When fitting the ConvLSTM model to the training data, how did you decide on the number of epochs (400), batch size (50), and learning rate? Did you choose these values through trial and error, or did you use a more structured approach like grid search or randomized search to find the best model and parameters? Also, was Early Stopping used to prevent overfitting by stopping training when the validation loss started increasing?

As mentioned in one of the questions before we were limited by the GPU-Memory of 40GB as well as how we plan to run the coupled model. For the number of timesteps that may be considered ahead, 30 days of atmospheric data is the upper limit as it is not feasible to store more than 30 daily atmospheric fields to predict river runoff one day ahead.
The hyperparameter testing itself was done by looping over a priori-defined set of parameters (More details are given below). The same applies to the batch size which was limited by computational constraints.

We did try using EarlyStopping but experienced some issues as the model often needed several epochs to improve. We decided to monitor the validation data MSE and save the best model weights based on this criterion. We simply used 400 epochs as all models' versions converged during this training duration. During the model's development we also implemented a new learning rate scheduler which addresses these plateaus during training but at this point we already removed the EarlyStopping. But your suggestions are interesting and we will implement EarlyStopping in the future again for training.

```
1.    callbacks = [
2.       ModelCheckpoint(
3.          dirpath="/silor/boergel/paper/runoff_prediction/data/modelWeights/",
4.          filename=f"{args.modelName}TopOne",
5.          save_top_k=1,
6.          mode="min",
7.          monitor="val_mse",
8.          save_last=True,
9.          ),
10.       PredictionPlottingCallback()
11.       ]
12.
```

- Additionally, it would help to explain why 30-day timesteps were chosen for the 4-channel atmospheric inputs, even though the runoff data is daily. This would make it clearer how the model is handling temporal input.

We added this information to the text.

L220: This window size allows the model to "remember" key atmospheric conditions leading up to a given day, enabling it to accurately predict runoff.

- Neural network hyperparameters section 3.4, you mention that "the model performance can be described as relatively robust when slightly changing the set of hyperparameters." Could you clarify what you mean by "slightly changing" the hyperparameters? It would be more helpful if you could quantify the model's performance for different sets of hyperparameter values to give a clearer picture of its robustness.

We looped over a predefined set of hyperparameters and evaluated their performance. For the sake of this review, we extended this loop to also consider smaller input sizes. Note that every iteration roughly takes 1 day.

```bash
1. #!/bin/bash
2.
3. HIDDEN_DIMS=(6 8 10)
4. KERNEL_SIZES=("(3,3)" "(5,5)" "(7,7)")
5. INPUT_SIZES=(10 20 30)
6. NUM_LAYERS=(1 2 3)
7.
8. for hidden_dim in "${HIDDEN_DIMS[@]}"; do
9.     for kernel_size in "${KERNEL_SIZES[@]}"; do
10.         for input_size in "${INPUT_SIZES[@]}"; do
11.             for num_layer in "${NUM_LAYERS[@]}"; do
12.
13.
model_name="ModelWeight_H${hidden_dim}_K${kernel_size//,/x}_I${input_size}_L${num_layer}"
14.
15.                 cmd="python trainForCoupledModel_more_sensitive.py --modelName $model_name --
hidden_dim $hidden_dim --kernel_size $kernel_size --input_size $input_size --num_layers
$num_layer --num_epochs 200"
16.
17.                 echo "Executing: $cmd"
18.                 $cmd
19.
20.             done
21.         done
22.     done
23. done
24.
```

To give the reader a better perspective of the model's robustness we added a comparison of the 10 "best" sets of parameters to the appendix.

All correlations and MAE's are comparable:

| | MAE | Correlation |
|---|---|---|
| 25 | <xarray.DataArray ()>\narray(141.11912972) | 0.999097 |
| 24 | <xarray.DataArray ()>\narray(145.20558121) | 0.999060 |
| 44 | <xarray.DataArray ()>\narray(151.68033482) | 0.998998 |
| 17 | <xarray.DataArray ()>\narray(148.73682357) | 0.998976 |
| 19 | <xarray.DataArray ()>\narray(154.97826802) | 0.998958 |
| 38 | <xarray.DataArray ()>\narray(167.9942031) | 0.998770 |

16  <xarray.DataArray ()>\narray(177.21611658)    0.998644

11  <xarray.DataArray ()>\narray(193.74296444)    0.998463

32  <xarray.DataArray ()>\narray(197.28541982)    0.998432

8   <xarray.DataArray ()>\narray(201.52035405)    0.998384

[Figure]

The best set of parameters was chosen and run for a longer time (more epochs, with a changing learning rate using cosine annealing)

We also added this Figure to the appendix:

[Figure]

L209-213: The model's performance can be described as relatively robust when changing the set of hyper parameters (see Figure \ref{fig-Supp3}. Interestingly, also shorter input sizes of 10 days perform really well. However, we still decided to use longer time scales, as we assume that longer input sizes increase the stability of the model needed for long-term climate simulations.

- In Section 4.1, where the ConvLSTM model's output is compared to the test data, Figure 5 shows the "total predicted river runoff." Could you clarify what exactly "total predicted river runoff" means? Does it represent the combined daily predicted runoff from all 97 rivers flowing into the Baltic Sea or for the four individual rivers? I'm assuming it's the total daily runoff for all these rivers, but confirming this would make things clearer.

You are right. We specified this in the text.

L228-229: The model's accuracy for the combined daily predicted runoff from all 97 rivers flowing into the Baltic Sea is displayed in Figure \ref{fig-error-metrics}.

- In Section 4.1, could you clarify why the E-HYPE data is labeled as the "original HYPE data" in the Figure 5 caption and in the results text, and in Figure 6's text and legend, it may be better to just refer to it as "E-HYPE data"? It would be clearer if it were consistently referred to as "E-HYPE data" throughout.

We agree and changed accordingly.

- Additionally, only a minor point to avoid confusion, it would help to consistently refer to "total predicted river runoff" instead of switching between "predicted river runoff" (as in line 215). "Original river runoff" should also be replaced with "E-HYPE data" for clarity.

  We agree and changed the text.

- In Section 4.1, it might be helpful to present the model performance results for the "total predicted river runoff" into the Baltic Sea in a table, showing metrics like accuracy, correlation, RMSE, and MAE. If available, showing these metrics individually for the four rivers—Neva, Oder, Umeälven, and Neman—would add useful detail, rather than only displaying residual errors for the daily runoff predictions on the four plots. Additionally, including density plots for the predictions of each of these four rivers could provide a clearer view of the model's performance for individual rivers.

  We added a table as well as a Figure showing the density plots to the appendix.

  L258: An overview of the individual error metrics is give in Table \ref{tab:error_metrics_rivers}.

  L210-213: The model's performance can be described as relatively robust when changing the set of hyper parameters (see Figure \ref{fig-Supp3}). Interestingly, also shorter input sizes of 10 days perform really well. However, we still decided to use longer time scales, as we assume that longer input sizes increase the stability of the model needed for long-term climate simulations.

- In Section 4.1, you make the point that the ConvLSTM model's performance is already satisfactory, the "discrepancies between the actual values and the predictions can partly be attributed to the use of a different atmospheric dataset than the ones originally used to drive the E-HYPE model". This is a key point, and it would be helpful to draw out this point earlier in the technical section when outlining the runoff data used for training and when describing the input datasets for the atmospheric forcing.

  We agree and added this information to the technical description in the sub-chapter 'Atmospheric Forcing'.
  L191: Lastly, it should be noted that UERRA is not the atmospheric dataset that was used to drive the original E-HYPE model.

- In Section 4.1 in Figure 5, you describe the right panel (b), showing the distribution of residuals as a density plot with a Gaussian shape—a bell curve centred around zero. You mention that there is no systematic bias, with residuals mostly within a narrow range around zero, though there is a slight positive bias at the peak. Could you explain this slight positive bias? Is it related to the differences in atmospheric datasets used in the ConvLSTM model versus those originally used as input in the E-HYPE model?

We think that this small shift is generated by the limited number of samples. More samples would likely reduce this bias. Moreover, the suggested sensitivity studies (Supplement Figure C1) support this assumption. However, on the other hand it is also possible that this small bias could be generated by the atmospheric dataset.

We changed the text accordingly.

L238-239: The bell-shaped curve is approximately centered around zero indicates that the model does not exhibit a systematic bias, […]

- In Figure 6, it might be helpful to use the same y-axis value range for all four plots showing the residual error. This would make it easier to see that Neva has the lowest residual error, with the ConvLSTM model's total predicted runoff within +/- 2.5%, compared to the other rivers.

  Changed.

- In Figure 6, the prediction errors are larger for the other three rivers compared to the River Neva. Have you considered whether this could be because the River Neva uses observed runoff data, while the other rivers rely on the E-HYPE hindcast simulation data? This could be linked to the issue of using different atmospheric datasets in the ConvLSTM model compared to the datasets originally used in the E-HYPE model.

  Thanks for pointing this out. Indeed, this is really interesting and was added to text.
  L251-253: Compared to the Neva River, the prediction errors are larger, which may be attributed to the training dataset, as the runoff for the Neva is based on measurements, whereas the other rivers are solely based on E-HYPE.

- In Section 4.1 in the Figure 6 caption, you mention that "the residuals were calculated as the relative difference between the predicted and observed values, normalized by the observed." However, the total runoff data is based on an E-HYPE hindcast simulation. Referring to "observed runoff data" makes it sound like this is measured runoff data from river gauges. Is this the case for all four rivers, or just for the Neva River? It would be helpful to clarify what the total predicted runoff data for each of the four individual rivers is compared to calculate the residual error.

  Changed in the text.

- In Figure A1, the legend refers to the "hydrological model," but it would be clearer to specify the E-HYPE model and ConvLSTM model. For Neva, the comparison should be with the measured flow data, as it is not based on the E-HYPE hindcast simulation data. Additionally, in Figure A1, you refer to the residuals as the relative difference between the predicted and observed values. However, these are not actually observed values but rather E-HYPE simulated values, except for Neva. It would be helpful to clarify this distinction.

  We agree and changed the figure accordingly.

- In Section 4.2, specifically in line 235, you mention that the predicted salinity from the ConvLSTM model matches the "original data" well, capturing short-term fluctuations effectively. It would be helpful to clarify what you mean by "original data"—is this the salinity forced with the E-HYPE runoff, or is it the measured salinity at BY15? In the Figure 7 legend, it would be clearer to use "ConvLSTM model" and "E-HYPE model" instead of "original". Additionally, in the caption, it might be better to avoid "original E-HYPE data" and simply use "E-HYPE data."

Changed accordingly.

- In Section 4.2, you don't address why the salinity at the surface, and to some extent at the bottom, as computed using the ConvLSTM model and E-HYPE runoff prediction with the MOM5 Ocean model, does not match well with the observed salinity at BY15. Specifically, at the surface it tends to over-predict the high and low salinity cycles. It would be helpful to acknowledge this discrepancy and offer some possible explanation for it.

We appreciate the reviewer's observation regarding the discrepancies between the simulated salinity and the observed values at BY15, particularly at the surface, where the high and low salinity cycles tend to be over-predicted. This behavior is not directly linked to the performance of the ConvLSTM river runoff model. Instead, it is attributed to the MOM5 ocean model's representation of physical processes, particularly the treatment of mixing, advection, and stratification in the Baltic Sea.

Several factors may contribute to this discrepancy:

Vertical Mixing and Stratification: The Baltic Sea is known for its strong vertical stratification due to the input of freshwater from rivers. The MOM5 model uses the K-profile parameterization (KPP) scheme for turbulence, which may not perfectly resolve small-scale mixing processes and vertical salinity gradients. This can result in an overestimation of salinity variability at the surface.

Lateral Advection and Exchange: While the MOM5 model captures the large-scale dynamics of the Baltic Sea, the lateral transport of saltwater from the Skagerrak into the central Baltic Sea may not be perfectly represented. This can introduce variability in surface and bottom salinity that is not observed in reality.

We emphasize that the focus of this study is on validating the river runoff prediction using the ConvLSTM model.

We added this information to the text as well.

L271-281: It should be noted that the discrepancies between the simulated salinity and the observed values at BY15 are not directly linked to the performance of the ConvLSTM river runoff model. Instead, it is attributed to the MOM5 ocean model's representation of physical processes, particularly the treatment of mixing, advection, and stratification in the Baltic Sea. Several factors may contribute to this discrepancy.

The Baltic Sea is known for its strong vertical stratification due to the input of freshwater from rivers. The MOM5 model uses the K-profile parameterization (KPP) scheme for turbulence, which may not perfectly resolve small-scale mixing processes and vertical salinity gradients. This can result in an overestimation of salinity variability at the surface. Moreover, while the MOM5 model captures the large-scale dynamics of the Baltic Sea, the lateral transport of saltwater from the Skagerrak into the central Baltic Sea may not be perfectly represented. This can introduce variability in surface and bottom salinity that are not observed in reality. However, all in all, the long-term trends and larger salinity changes are accurately captured, indicating the model's robustness in predicting high-frequency and low-frequency variations.

- In Section 5, you conclude that all results lie within the error margin of the hydrological model itself when compared to observations, with the average error on daily time scales for individual rivers mostly under 10%. It would be helpful to mention this average error of 10% earlier in Section 4.1, specifically around line 255, when introducing supplementary Figure A1. This would provide context for the reader before the conclusion in Section 5.

  We agree and added this information to the text.

- In Section 5, you conclude that the ConvLSTM model is significantly less computationally intensive than running comprehensive hydrological models. Could you provide a more detailed quantification of the reduction in computational demand when forecasting with the ConvLSTM model compared to these hydrological models? Have you tested the computing speed against any other traditional hydrological modes or only the E-HYPE hydrological model to make this conclusion?

  We agree that this information is useful and added it to the text. Our model generates one year of daily river runoff in roughly 10 seconds. The runtime of a hydrological model (personal communication with Stefan Hagemann (Dr. Stefan Hagemann, Regional Land and Atmosphere Modeling, Head of Department), with a similar resolution varies between 5-15 minutes per year.

  This results in a speed up in the range of factor 30-90.

  L309-310: The achieved speedup (depending on the complexity of the hydrological model) is within the range of 30 to 90 times faster.

Technical Corrections:
Other minor suggestions:
- The title could be simplified by changing "leveraging" to "using" and removing "comprehensive" before "discharge forecasting." You could also add "into the Baltic Sea" for more clarity.
  Changed.
- The abstract is currently too broad, and it would benefit from including some specific numerical results to quantify the ConvLSTM model's performance against the e-HYPE

data and compared to traditional hydrological models. This would demonstrate the ConvLSTM model's effectiveness at predicting runoff.

Changed.

- Equation 1 line 60 for Xtk the k I think should be subscript.

    Changed.

- Overall, the study presents a novel approach to forecasting river runoff using ConvLSTM network models. The ConvLSTM model performs similarly to traditional hydrological models such as the E-HYPE model in capturing runoff patterns but offers faster processing and greater computational efficiency, which makes it a valuable contribution to the field. However, I think some details need further explanation, and in places more clarity is required. Minor revisions would strengthen the paper, but this use of ConvLSTM models to forecast runoff on such a widespread scale of the Baltic Sea catchment is definitely of interest to others in the field and a excellent study.

    Thank you very much for your detailed review, which improved the current version of the article a lot.